# Characteristics of Intestinal Barrier State and Immunoglobulin-Bound Fraction of Stool Microbiota in Advanced Melanoma Patients Undergoing Anti-PD-1 Therapy

**DOI:** 10.3390/ijms26168063

**Published:** 2025-08-20

**Authors:** Bernadeta Drymel, Katarzyna Tomela, Łukasz Galus, Agnieszka Olejnik-Schmidt, Jacek Mackiewicz, Mariusz Kaczmarek, Andrzej Mackiewicz, Marcin Schmidt

**Affiliations:** 1Department of Biotechnology and Food Microbiology, Poznań University of Life Sciences, 60-627 Poznań, Poland; agnieszka.olejnik-schmidt@up.poznan.pl; 2Department of Cancer Immunology, Poznań University of Medical Sciences, 61-866 Poznań, Poland; ktomela@gmail.com (K.T.); mariusz.kaczmarek@wco.pl (M.K.); andrzej.mackiewicz@wco.pl (A.M.); 3Department of Medical and Experimental Oncology, Institute of Oncology, Poznań University of Medical Sciences, 60-355 Poznań, Poland; lukasz_galus@wp.pl (Ł.G.); jmackiewicz@ump.edu.pl (J.M.); 4Department of Cancer Diagnostics and Immunology, Greater Poland Cancer Centre, 61-866 Poznań, Poland

**Keywords:** intestinal barrier, secretory immunoglobulin A, gut microbiota, advanced melanoma, immune checkpoint inhibitors

## Abstract

The gut microbiota is recognized as one of the extrinsic factors that modulate the clinical outcomes of immune checkpoint inhibitors (ICIs), such as inhibitors targeting programmed cell death protein 1 (PD-1), in cancer patients. However, the link between intestinal barrier, which mutually interacts with the gut microbiota, and therapeutic effects has not been extensively studied so far. Therefore, the primary goal of this study was to investigate the relationship between intestinal barrier functionality and clinical outcomes of anti-PD-1 therapy in patients with advanced melanoma. Fecal samples were collected from 64 patients before and during anti-PD-1 therapy. The levels of zonulin, calprotectin, and secretory immunoglobulin A (SIgA), which reflect intestinal permeability, inflammation, and immunity, respectively, were measured in fecal samples (*n* = 115) using an Enzyme-Linked Immunosorbent Assay (ELISA). Moreover, the composition of the immunoglobulin (Ig)-bound (*n* = 108) and total stool microbiota (*n* = 117) was determined by the V3–V4 region of 16S rRNA gene sequencing. ELISA indicated a higher baseline concentration of fecal SIgA in patients with favorable clinical outcomes than those with unfavorable ones. Moreover, high baseline concentrations of intestinal barrier state biomarkers correlated with survival outcomes. In the cases of fecal zonulin and fecal SIgA, there was a positive correlation, while in the case of fecal calprotectin, there was a negative correlation. Furthermore, there were differences in the microbial profiles of the Ig-bound stool microbiota between patients with favorable and unfavorable clinical outcomes and their changes during treatment. Collectively, these findings indicate an association between intestinal barrier functionality and clinical outcomes of anti-PD-1 therapy in advanced melanoma patients.

## 1. Introduction

The association between the gut microbiota composition and clinical outcomes of immune checkpoint inhibitors (ICIs), such as inhibitors targeting cytotoxic T lymphocyte antigen-4 (CTLA-4), programmed cell death protein 1 (PD-1), and its ligand (PD-L1), was demonstrated in various cancer cohorts [1,2,3,4,5]. Although the mechanism of gut microbiota-mediated regulation of response to ICIs is not fully understood, the gut microbiota is known to exert a critical impact on local and systemic immunity. It constantly interacts with other components of the intestinal barrier, i.e., the mucus layer, epithelial cells, and lamina propria—a connective tissue containing the gut-associated lymphoid tissue (the largest component of the immune system) [6,7]. In normal physiological conditions, this interaction provides a homeostatic relationship between the intestinal microorganisms and the host. However, the loss of barrier integrity may lead to uncontrolled luminal antigen translocation to the host internal environment and trigger local and systemic inflammation, which, in cancer patients, may affect immune responses at the tumor site and the clinical outcomes of the ICIs.

Despite growing evidence suggesting an involvement of the gut microbiota in shaping immune response in cancer patients undergoing ICI therapy, the intestinal barrier state and mucosal immune system activity in those patients are poorly explored. To date, a limited number of studies have reported the differences in the intestinal barrier function between patients with favorable and unfavorable outcomes of ICIs. For instance, Ouaknine Krief et al. (2019) found that high baseline citrulline level (≥20 μM) in blood plasma, which reflects proper intestinal epithelial cell function, was associated with better response and clinical benefit to the ICI therapy and longer progression-free survival (PFS) and overall survival (OS) in patients with advanced non-small cell lung cancer (NSCLC) receiving the anti-PD-1 therapy [8]. Another study revealed a higher abundance of inflammatory cells, such as dendritic cells, monocytes, macrophages, and neutrophils, in fecal samples of progressors than non-progressors in a cohort of melanoma patients receiving the anti-PD-1 antibodies, which implies increased activation of inflammatory immune responses within the intestinal mucosa in the subgroup with poor clinical outcomes [2].

The primary goal of the present study was to evaluate the intestinal barrier functionality in advanced melanoma patients undergoing anti-PD-1 therapy and analyze its association with the clinical outcomes of immunotherapy. Therefore, the concentration of zonulin, calprotectin, and secretory immunoglobulin A (SIgA) was measured using Enzyme-Linked Immunosorbent Assay (ELISA) in stool samples collected from those patients before the start (at baseline) and during treatment. Zonulin is considered a biomarker of intestinal permeability. It is known as a physiological modulator of intercellular tight junctions (TJs) that are localized near the apical surface of adjacent epithelial cells and control the passage of luminal antigens via the paracellular pathway. Zonulin binding to a specific receptor on the surface of intestinal epithelium activates a cascade of biochemical events that induce TJ disassembly, followed by increased permeability (increased passage of luminal antigens) and activation of mucosal immunity. The zonulin pathway is part of a defensive mechanism that initiates immune responses against antigens to remove them from the microbial ecosystem. Therefore, it may play an important role in maintaining homeostasis in the intestinal mucosa. However, disruption of intestinal barrier integrity (reflected by increased zonulin levels) may also contribute to the development of various chronic inflammatory diseases [9,10,11]. On the other hand, fecal calprotectin is a biomarker reflecting intestinal inflammation. It belongs to the family of calcium-binding proteins constitutively expressed in neutrophils and other cells that regulate inflammatory processes and exhibit antibacterial and antiproliferative activity. During intestinal inflammation, neutrophils are recruited to the intestinal mucosa, and their numbers positively correlate with the concentration of fecal calprotectin [12]. Finally, fecal SIgA is regarded as a biomarker of intestinal immunity. It is the dominant Ig in the mucosal secretions and plays a crucial role in maintaining homeostasis in the intestinal mucosa. Firstly, it protects the intestinal epithelium against pathogens through the process known as ‘immune exclusion’. SIgA stimulates pathogen cross-linking and entrapment in the intestinal lumen, which facilitates their clearance from the gastrointestinal tract through peristalsis. Secondly, SIgA provides homeostasis in the intestinal mucosa through the interaction with commensal microbiota. On the one hand, SIgA coating of non-pathogenic species may prevent their invasion into the intestinal mucosa. On the other hand, it may promote intestinal colonization by commensal species by supporting their adhesion to the mucus layer or acting as a nutritional source [13,14]. Furthermore, SIgA was shown as a key component of the gut environment that shapes the oral tolerance to food proteins, potentially by preventing their translocation through the gut barrier [15].

In the present study, ELISA results indicated that the baseline concentration of fecal SIgA was significantly higher in patients with favorable clinical outcomes than those with unfavorable ones. Moreover, the analyzed biomarkers were elevated in most of the patients (concerning the reference ranges [9,16,17,18]). High baseline concentrations of intestinal barrier state biomarkers significantly correlated with survival outcomes; in the cases of fecal zonulin and fecal SIgA, it was a positive correlation, while in the case of fecal calprotectin, a negative correlation. Taking into account that fecal SIgA level was associated with improved clinical outcomes in the study cohort, the fraction of stool microbiota coated with immunoglobulins (Igs) was purified from the total stool microbiota, and bacterial composition of the Ig-bound stool microbiota fraction and total stool microbiota was characterized using V3–V4 region of 16SrRNA gene sequencing. It was found that the Ig-bound stool microbiota shared a substantial subset of genera with the total stool microbiota, however, at different abundances. Moreover, there were differences in the microbial profiles of the Ig-bound stool microbiota between patients with favorable and unfavorable clinical outcomes and their changes during treatment. Taken together, these findings demonstrated the association between the intestinal barrier state and clinical outcomes of anti-PD-1 therapy in the study cohort. This suggest that the functionality of the intestinal barrier and immunity may affect the whole-body immune system, and therefore, play a role in shaping the anti-cancer immune responses to immunotherapy or reflect the overall immunological status of patients and ability of their immune system to fight cancer.

## 2. Results

### 2.1. Intestinal Barrier State Biomarkers

Baseline characteristics of advanced melanoma patients undergoing anti-PD-1 therapy were described in our previous paper [3]. Briefly, there were no statistically significant differences in terms of age, sex, and distribution of the metastatic stage and serum lactate dehydrogenase (LDH) level between patient subgroups (*p*-value > 0.05). However, the median value of serum LDH was significantly higher (*p* ≤ 0.05) in the non-responders (NR) compared to the responders (R).

To evaluate the intestinal barrier functionality in the study cohort, the concentration of zonulin, calprotectin, and SIgA in stool samples (n = 115) was measured using ELISA kits.

#### 2.1.1. A High Baseline Concentration of Fecal SIgA Is Associated with Response and Clinical Benefit from the Anti-PD-1 Therapy

In this study, there were no statistically significant differences in the fecal zonulin and calprotectin concentrations between R vs. NR and patients, who clinically benefited from the anti-PD-1 therapy (CB) vs. those, who did not (NB) at baseline and during the therapy (*p*-value > 0.05, Figure 1A–D). However, the baseline concentration of fecal SIgA was significantly higher in patients with favorable clinical outcomes than those with unfavorable ones (*p*-value ≤ 0.05, R0 vs. NR0 and CB0 vs. NB0, median values, respectively, 5237 vs. 2539 and 4324 vs. 1671 µg·mL^−1^, Figure 1E,F). The biomarker levels have not changed significantly during treatment within subgroups with favorable and unfavorable clinical outcomes (*p*-value > 0.05, Figure 1A–F). These findings indicate a lack of profound influence of the anti-PD-1 therapy on the intestinal barrier functionality. However, the baseline difference in the fecal SIgA concentrations underlines an increased activation of intestinal immunity in cancer patients with a favorable outcomes of the immunotherapy.

#### 2.1.2. Elevated Levels of Intestinal Barrier State Biomarkers in Advanced Melanoma Patients

The concentrations of all investigated biomarkers were increased in most of the samples according to the reference ranges indicated by the ELISA kit manufacturer [16,17]. In detail, elevated levels of fecal SIgA (>2040 µg·mL^−1^) were found in 64%, normal (510–2040 µg·mL^−1^)—in 35%, and subnormal (<510 µg·mL^−1^)—in 1% of all analyzed samples (i.e., samples collected from all patients regardless of clinical outcome and collection time point). Similarly, calprotectin concentrations were borderline elevated or elevated (≥50 µg·g^−1^) in 62% of samples. Regarding fecal zonulin, the manufacturer did not indicate the reference range. Therefore, the results were compared to the median level in healthy individuals (61 ng·mL^−1^) as presented in the manufacturer’s guideline [18] and the cut-off point used by Jendraszak et al. [9], where a fecal zonulin level > 105 ng·mL^−1^ was regarded as elevated. The fecal zonulin concentrations exceeded those values in 86% and 64% of samples, respectively.

Similar trends were observed in the study subgroups at baseline. Briefly, in the R and CB subgroups, baseline levels of fecal SIgA were elevated in 63% and 68% of patients, respectively, fecal calprotectin in 58% and 59%, respectively, and fecal zonulin (>105 ng·mL^−1^) in 71% of patients in both subgroups. In the NR and NB subgroups, 58% and 68% of patients had increased levels of fecal SIgA at baseline, 61% and 62% had elevated levels of fecal calprotectin, and 68% and 67% had increased levels of fecal zonulin, respectively. It is worth noting that a subnormal level of baseline fecal SIgA was only detected in one patient with unfavorable clinical outcomes.

At T_1_, intestinal barrier state biomarkers were also elevated in most of the patients with favorable clinical outcomes. In the R and CB subgroups, approximately 80% of patients had elevated concentrations of fecal SIgA (>2040 µg·mL^−1^ in 78% and 81% of patients, respectively). Moreover, fecal calprotectin levels (≥50 µg·g^−1^) were increased in 65% of patients in both subgroups, and fecal zonulin concentrations (>105 ng·mL^−1^) were raised in 70% and 65% of patients, respectively. In the subgroups of patients with unfavorable clinical outcomes, these trends were more diverse. In the NR subgroup, fecal zonulin, fecal calprotectin, and fecal SIgA levels were increased in 59%, 71%, and 65% of patients, respectively. For comparison, in the NB subgroup, the concentrations of fecal zonulin and fecal calprotectin were also elevated in most patients (67% and 78%, respectively), while fecal SIgA levels were increased only in 44% of patients.

These results demonstrated increased intestinal barrier permeability, inflammation, and immunity in most advanced melanoma patients before and during anti-PD-1 therapy, suggesting impaired functioning of the intestinal barrier. However, it is worth noting that fecal SIgA levels were in the normal range in most patients with disease progression (PD) at T_1_, while in most patients who benefited from the therapy, they were elevated. This implies an association between enhanced Ig responses in the gut and improved clinical outcomes of the anti-PD-1 therapy.

#### 2.1.3. Mutual Correlations Between Intestinal Barrier State Biomarkers and Total Stool Microbiota Composition

Elevated concentrations of analyzed biomarkers (according to reference ranges [9,16,17,18]) indicated some disturbances in the functioning of the intestinal barrier in the study cohort. As there might be a link between disrupted host-microbe interaction that may lead to increased intestinal permeability, which, in turn, may contribute to gut inflammation and induce mucosal immune responses, the mutual correlations between intestinal barrier state biomarkers were investigated in this study. Moreover, their associations with the *Firmicutes* to *Bacteroidetes* (F/B) ratio in the total stool microbiota (disruptions in the biomarker value may reflect gut microbiota dysbiosis that links with the development of various diseases [19,20]) were analyzed.

In all analyzed samples, fecal SIgA negatively correlated with the F/B ratio in the total stool microbiota (*p*-value = 0.026, Figure 2A). A similar trend was found in patients with unfavorable clinical outcomes at baseline (*p*-value ≤ 0.05, Figure 2D,E) and in the NB subgroup at T_1_ (R = −0.68, *p*-value = 0.05, Appendix A).

In contrast, fecal calprotectin positively correlated with the F/B ratio in the total stool microbiota and fecal zonulin in all analyzed samples (*p*-value ≤ 0.05, Figure 2B,C). Comparable trends were observed in patients with favorable clinical outcomes. In detail, in the R and CB subgroups, there was a positive correlation between fecal calprotectin and fecal zonulin at baseline (*p*-value ≤ 0.05, Figure 2H,I), whereas in the R subgroup, fecal calprotectin positively correlated with the F/B ratio in the total stool microbiota at T_0_ and T_1_ (*p*-value ≤ 0.05, Figure 2F,G).

Distinct correlations between analyzed biomarkers in advanced melanoma patients with favorable vs. unfavorable clinical outcomes of anti-PD-1 therapy suggest activation of different mechanisms within the intestinal mucosa in those subgroups that may affect or mirror whole-body immunity and influence immunotherapy efficacy. The list presenting Spearman’s ρ values and *p*-values describing their statistical significance calculated for analyzed biomarkers in all samples (for general trend observation) and study subgroups is available in Appendix A.

#### 2.1.4. Baseline Levels of Intestinal Barrier State Biomarkers Are Associated with the Survival Outcomes

The association between baseline concentrations of intestinal barrier state biomarkers and survival outcomes (expressed as PFS and OS) was also examined in this study. Survival analysis indicated that high fecal zonulin and SIgA levels at baseline predicted improved survival outcomes, while high baseline fecal calprotectin levels—the poor ones (Figure 3).

Specifically, there was a significant association between baseline zonulin level and OS probability (*p*-value = 0.014, Figure 3C), and the concentration of the biomarker higher than 303 ng·mL^−1^ was significantly associated with decreased risk of death (OS hazard ratio (HR) 0.29 (0.099–0.83), *p*-value = 0.021, Figure 3D). The median baseline concentrations of fecal zonulin in the R and CB subgroups (281 and 216 ng·mL^−1^, respectively) were comparable to this predicting longer OS, whereas, in the NR and NB subgroups, the median levels of fecal zonulin at baseline were 160 ng·mL^−1^ in both subgroups, i.e., ~2 times lower than the estimated cut-off point.

Fecal SIgA level at baseline was also significantly associated with PFS and OS probability (*p*-value ≤ 0.05, Figure 3I,K). The baseline concentration of fecal SIgA higher than 820 µg·mL^−1^ was significantly associated with decreased risk of disease progression (PFS HR 0.34 (0.14–0.85), *p*-value = 0.021, Figure 3J) and higher than 5477 µg·mL^−1^ with decreased risk of death (OS HR 0.27 (0.093–0.78), *p*-value = 0.015, Figure 3L). For comparison, the median concentrations of fecal SIgA in the R and CB subgroups at T_0_ (5237 and 4324 µg·mL^−1^, respectively) were ~6 and ~5 times higher, respectively, than the concentration that predicted longer PFS and comparable to the concentration predicting a longer OS. In the NR and NB subgroups, the median concentrations of fecal SIgA at T_0_ (2539 and 1671 µg·mL^−1^, respectively) were ~3 and ~2 times higher, respectively, than the concentration associated with longer PFS, whereas ~2 and ~3 times lower, respectively, than the concentration that predicted longer OS.

Moreover, there was a significant association between baseline fecal calprotectin level and PFS and OS probability (*p*-value ≤ 0.05, Figure 3E,G). The baseline concentration of the biomarker higher than 212 µg·mL^−1^ was significantly associated with increased risk of disease progression (PFS HR 2.5 (1–6), *p*-value = 0.042, Figure 3F), and there was a borderline significant association with increased risk of death (OS HR 2.6 (0.98–7.1), *p*-value = 0.054, Figure 3H). For comparison, the median concentrations of fecal calprotectin in the R and CB subgroups at T_0_ were ~4 times lower than the concentration predicting shorter PFS and OS (R0 and CB0, median values, respectively, 55 and 59 µg·mL^−1^), while in NR and NB, they were ~3 and ~2 times lower, respectively (NR0 and NB0, median values, respectively, 77 and 98 µg·mL^−1^).

Collectively, the association between the baseline biomarker levels and survival outcomes suggests that intestinal barrier functionality (that mutually interacts with the gut microbiota to provide its homeostatic relationship with the host) may contribute to shaping anti-cancer immune responses or demonstrate the activation of the immune system against cancer. This also implies that baseline intestinal barrier state biomarkers may serve as prognostic biomarkers in advanced melanoma patients enrolled in the treatment with ICIs.

### 2.2. Ig-Bound Stool Microbiota Fraction

SIgA was found to contribute to intestinal homeostasis and regulate the gut microbiota composition [13,14]. In this study, the median concentration of fecal SIgA at baseline was higher in patients with favorable clinical outcomes than those with unfavorable ones (Figure 1E,F), and an increased level of the biomarker was associated with improved survival outcomes (Figure 3I–L). These findings suggest that intestinal immunity may influence the anti-cancer effects of the anti-PD-1 therapy in advanced melanoma patients, potentially through the interaction with the gut microbiota. Taking this into account, in the present study, the Ig-bound stool microbiota fraction was purified from the total stool microbiota, and composition of the Ig-bound stool microbiota (n = 108) and the total stool microbiota (n = 117) was characterized using V3–V4 hypervariable regions of 16S rRNA gene sequencing.

#### 2.2.1. The Dominance of *Bacillota* Phylum in the Ig-Bound Stool Microbiota and Changes in the Taxonomic Profile at the Phylum Level over the Study Period

At the phylum level, the Ig-bound stool microbiota was dominated by phylum *Bacillota* (formerly *Firmicutes*) in all study subgroups (its relative abundance ranged from 78% to 89% in R/NR subgroups and from 76% to 93% in CB/NB subgroups, Figure 4B,D). For comparison, the total stool microbiota was dominated by phyla *Bacillota* and *Bacteroidota* (formerly *Bacteroidetes*) in all study subgroups (Figure 4A,C). However, the dominance of phylum *Bacillota* was less remarkable than in the Ig-bound stool microbiota (its relative abundance ranged from 57% to 64% in R/NR subgroups and from 58% to 62% in CB/NB subgroups, Figure 4A,C).

It is worth noting that the relative abundances of particular phyla in the Ig-bound stool microbiota have changed significantly during the anti-PD-1 therapy. Specifically, in the NR subgroup, the relative abundances of phyla *Bacteroidota* and *Pseudomonadota* (formerly *Proteobacteria*) decreased during treatment (*p*-value ≤ 0.05, NR0 (9.3%) vs. NR1 (3.7%), NR0 (9.3%) vs. NRn (2.9%), Figure 5A; NR0 (4.7%) vs. NRn (0.4%), Figure 5C). In contrast, the relative abundance of phylum *Bacillota* increased in the NR subgroup at T_1_ and was significantly higher than at T_0_ (*p*-value ≤ 0.05, NR0 (77.9%) vs. NR1 (88.5%), Figure 5D). In patients with favorable clinical outcomes, i.e., in the R and CB subgroups, there was a decrease in the relative abundance of phylum *Bacteroidota* at T_n_ (*p*-value ≤ 0.05, R1 (7.5%) vs. Rn (2.5%), CB0 (6.2%) vs. CBn (2.7%), and CB1 (6.4%) vs. CBn (2.7%), Figure 5A,B). Moreover, at T_1_, the Ig-bound stool microbiota of the CB subgroup was significantly enriched in phylum *Verrucomicrobiota* (formerly *Verrucomicrobia*) members as compared to the NB subgroup (*p*-value ≤ 0.01, CB1 (2.4%) vs. NB1 (0.2%), Figure 5E). Similarly to trends observed in the Ig-bound stool microbiota, the relative abundance of phylum *Pseudomonadota* in the total stool microbiota decreased in the NR subgroup during treatment (*p*-value ≤ 0.0001, NR0 (5.1%) vs. NRn (1.2%), Appendix A), and the total stool microbiota of the CB subgroup was also enriched in phylum *Verrucomicrobiota* at T_1_ (*p*-value ≤ 0.001, CB1 (1.2%) vs. NB1 (0.1%), Appendix A). Other statistically significant differences in the relative abundances of particular phyla in the total stool microbiota between study subgroups that were not consistent with those observed in the Ig-bound stool microbiota are shown in Appendix A.

Taken together, the composition of the Ig-bound stool microbiota at the phylum level differed from the total stool microbiota composition, revealing that Igs targeted a specific subset of microbial taxa. Moreover, the abundance patterns at the phylum level in the Ig-bound stool microbiota have changed during anti-PD-1 therapy. Noteworthy, more remarkable alternations were found in the NR subgroup vs. patients with favorable clinical outcomes, suggesting that Ig response modification, potentially associated with the introduction of anti-PD-1 antibodies, could negatively affect clinical outcomes of the immunotherapy or reflect unfavorable changes in the whole-body immunity that lead to poor therapeutic effects.

#### 2.2.2. Differences in the Ig-Bound Stool Microbiota Signatures Between Patients with Favorable and Unfavorable Clinical Outcomes at Baseline (at T_0_) and During the Anti-PD-1 Therapy (at T_1_ and T_n_)

The differential abundance analysis (DAA) was performed at the genus level to determine differentially abundant genera in the Ig-bound stool microbiota between patients with favorable and unfavorable clinical outcomes of the anti-PD-1 therapy before and after its initiation. Statistically significant results (*p*-value < 0.1) of the DAA are presented in Figure 6.

At baseline, the DAA indicated 11 differentially abundant genera in the R vs. NR comparison and 10 in CB vs. NB (*p*-value < 0.1, Figure 6). Among those with the highest statistical significance (*p*-value ≤ 0.05), there was a higher relative abundance of genus *Dorea* in the R subgroup. In contrast, in patients with unfavorable clinical outcomes, i.e., in the NR and/or NB subgroup, there was an enrichment in genera *Colidextribacter*, *Intestinimonas*, *Ruminococcaceae* Incertae Sedis, bacterium from family *Ruminococcaceae*, *Desulfovibrio*, *Escherichia-Shigella*, *Enterococcus*, *Odoribacter*, *Citrobacter*, and bacterium from family *Enterobacteriaceae* (*p*-value ≤ 0.05). Consistently, a comparable number of differentially abundant genera was indicated in the comparison of the total stool microbiota signatures between those subgroups at baseline (Appendix A). Several genera that were more relatively abundant in the Ig-bound stool microbiota of patients with unfavorable clinical outcomes (*p*-value < 0.1, Figure 6) were also enriched in their total stool microbiota, such as genera *Christensenellaceae* R-7 group, *Intestinimonas*, bacterium from family *Ruminococcaceae*, *Enterococcus*, and *Ruminococcaceae* Incertae Sedis (*p*-value < 0.1, Appendix A).

At T_1_, 2 and 6 differentially abundant genera were found in the comparison of R vs. NR and CB vs. NB, respectively (*p*-value < 0.1, Figure 6). Among those with the highest statistical significance (*p*-value ≤ 0.05), the genus *Akkermansia* was enriched in the R and CB subgroups. It is worth noting that its higher relative abundance was also indicated in the total stool microbiota of those subgroups at T_1_ (*p*-value ≤ 0.05, Appendix A). The genus *Akkermansia* is a member of the phylum *Verrucomicrobiota*, which was also significantly enriched in the Ig-bound and total stool microbiota of the CB vs. NB subgroup at T_1_ (Figure 5E and Appendix A). Moreover, in the CB subgroup at T_1_, there was a higher relative abundance of genus *Christensenellaceae* R-7 group in the Ig-bound and total stool microbiota (*p*-value < 0.1, Figure 6B and Appendix A). On the other hand, genera *Intestinibacter* and *Holdemanella* were enriched in the Ig-bound and total stool microbiota of the NB subgroup at T_1_ (*p*-value < 0.1, Figure 6B and Appendix A). Noteworthy, the DAA indicated more differentially abundant genera in the total stool microbiota (i.e., 8 and 13 in the R vs. NR and CB vs. NB comparison, respectively, *p*-value < 0.1, Appendix A) than in the Ig-bound stool microbiota fraction between those subgroups at T_1_.

At T_n_, there were 5 differentially abundant genera in the comparison between the R and NR subgroups (for comparison, 4 differentially abundant genera were indicated in the total stool microbiota between those subgroups, *p*-value < 0.1, Figure 6 and Appendix A). Among those with the highest statistical significance (*p*-value ≤ 0.05), there was a higher relative abundance of genus [*Eubacterium*] *hallii* group in the R subgroup and genus [*Eubacterium*] *ventriosum* group in the NR subgroup. Interestingly, in the total stool microbiota, the genus [*Eubacterium*] *hallii* group was enriched in the NR subgroup at T_n_ (*p*-value ≤ 0.05, Appendix A). On the other hand, there was an enrichment in the genus *Fusicatenibacter* in the Ig-bound stool microbiota of the NR subgroup at T_n_ (*p*-value < 0.1), which was also more relatively abundant in the total stool microbiota of the subgroup (*p*-value ≤ 0.05, Appendix A). Due to the low sample size in the NB subgroup, the comparison of the Ig-bound stool microbiota and total stool microbiota signatures between the CB and NB subgroups at T_n_ was not performed.

The comparison of microbial signatures in the Ig-bound stool microbiota between subgroups of patients with distinct clinical outcomes indicated the highest number of differentially abundant genera at baseline. In contrast, the differences at T_1_ and T_n_ between those subgroups were less noticeable. In line with that, there was also a difference in the baseline fecal SIgA concentrations between those subgroups (Figure 1). Together, these findings indicated association between baseline intestinal immunity and the clinical outcomes of the anti-PD-1 therapy.

#### 2.2.3. Changes in the Ig-Bound Stool Microbiota Signatures During the Anti-PD-1 Therapy in Patients with Favorable and Unfavorable Clinical Outcomes (T_0_ vs. T_1_, T_0_ vs. T_n_, and T_1_ vs. T_n_)

The DAA was also performed to analyze the changes in the relative abundances of genera in the Ig-bound stool microbiota during the anti-PD-1 therapy. The microbial signatures of the Ig-bound stool microbiota at particular collection time points (T_0_ vs. T_1_, T_0_ vs. T_n_, and T_1_ vs. T_n_) were compared within subgroups with favorable and unfavorable clinical outcomes. Statistically significant results (*p*-value < 0.1) of the DAA are presented in Figure 6.

In patients with favorable clinical outcomes, there were found few statistically significant changes in the relative abundances of genera during the anti-PD-1 therapy (*p*-value < 0.1, Figure 6A,B). The relative abundances of genera *Bacteroides*, *Citrobacter* (T_0_ vs. T_n_ and T_1_ vs. T_n_), and bacterium from family *Enterobacteriaceae* (T_0_ vs. T_n_) were decreased at T_n_ in the R and CB subgroups (*p*-value < 0.1). In contrast, at T_n_, enrichment in the genus *Oscillospiraceae* NK4A214 group was found in the R subgroup (T_0_ vs. T_n_), and in the genus *Romboutsia* in the R (T_1_ vs. T_n_) and CB (T_0_ vs. T_n_) subgroups (*p*-value < 0.1). Moreover, there was also a reduction in the relative abundance of genus *Clostridium* sensu stricto 1at T_1_ (T_0_ vs. T_1_) in the R subgroup and an enrichment in this genus at T_n_ (T_1_ vs. T_n_) in the R and CB subgroups (*p*-value < 0.1). Accordingly, there was an increase in the relative abundances of genera *Clostridium* sensu stricto 1 and *Romboutsia* in the total stool microbiota of patients with favorable clinical outcomes at T_n_ (*p*-value < 0.1, Appendix A). Similarly to trends observed in the Ig-bound stool microbiota signatures, there were few statistically significant changes in the relative abundances of genera in the total stool microbiota during treatment in patients with favorable clinical outcomes (*p*-value < 0.1, Appendix A).

In contrast, in patients with unfavorable clinical outcomes, the DAA indicated 7 and 8 differentially abundant genera in the NR and NB subgroups, respectively, in the comparison of microbial signatures at T_0_ vs. T_1_ (*p*-value < 0.1). A decrease in the relative abundances of genera *Odoribacter*, bacterium from family *Ruminococcaceae*, and *Citrobacter* and an increase in the genus *Dorea* was found in the NR and NB subgroups at T_1_ (*p*-value < 0.1). Moreover, among taxa with the highest statistical significance (*p*-value ≤ 0.05), there was also a reduction in the relative abundance of genus *Butyricimonas* in the NR subgroup, and an enrichment in genera *Romboutsia* and *Agathobacter* in the NR and NB subgroups, respectively, at T_1_. For comparison, in the total stool microbiota, the relative abundance of the bacterium from the family *Ruminococcaceae* also decreased in the NR subgroup at T_1_ (*p*-value < 0.1, Appendix A). Moreover, similarly to trends observed in the Ig-bound stool microbiota of the NB subgroup (*p*-value < 0.1), there was a decrease in the relative abundance of genus *Christensenellaceae* R-7 group and an increase in genus *Intestinibacter* in the total stool microbiota at T_1_ both in the NR and NB subgroups (*p*-value < 0.1, Appendix A).

There were also found 7 and 3 differentially abundant genera in the comparison of microbial signatures at T_0_ vs. T_n_ and T_1_ vs. T_n_ in the NR subgroup, respectively (*p*-value < 0.1, Figure 6A). At T_n_, there was a decrease in the relative abundance of the genus *Lachnospiraceae* NK4A136 group and an increase in genera *Anaerostipes* and [*Eubacterium*] *ventriosum* group (T_0_ vs. T_n_ and T_1_ vs. T_n_, *p*-value < 0.1). Moreover, in the comparison of microbial signatures at T_0_ vs. T_n_, there was also a decrease in genera *Odoribacter* and *Citrobacter*, among the results with the highest statistical significance (*p*-value ≤ 0.05). An enrichment in the genus *Anaerostipes* during treatment was also observed in the total stool microbiota in the NR subgroup (T_0_ vs. T_n_ and T_1_ vs. T_n_, *p*-value ≤ 0.05, Appendix A). Due to the low sample size in the NB subgroup at T_n_, the DAA was not performed to analyze changes in the Ig-bound stool microbiota and total stool microbiota signatures during the anti-PD-1 therapy within the NB subgroup (T_0_ vs. T_n_ and T_1_ vs. T_n_).

Collectively, the microbial abundance patterns at the genus level in the Ig-bound stool microbiota have changed during anti-PD-1 therapy in advanced melanoma patients. However, these alterations were more remarkable in the subgroup with unfavorable clinical outcomes, consistently with trends observed at the phylum level (Figure 5). These findings revealed changes in the specificity of Ig responses that could be a cause or effect of alterations in the systemic immunity that lead to unfavorable outcomes of the immunotherapy.

#### 2.2.4. Comparison Between the Ig-Bound and Total Stool Microbiota Composition

Beta diversity analysis with principal coordinates analysis (PCoA) performed for comparison of the Ig-bound and total stool microbiota indicated that the microbiotas shared a substantial subset of bacterial genera (Figure 7A; as indicated while using weighted-UniFrac) but at different abundances (Figure 7B; as indicated while using unweighted-UniFrac). The circulating cell-free microbial DNA (cfmDNA) extracted from the blood plasma samples of the study cohort (published in our previous paper [21]) was included in the plot to better visualize the relations between microbiotas (Figure 7A,B).

As it was indicated in the beta diversity analysis, a substantial subset of bacterial genera was found both in the Ig-bound and total stool microbiota (from 53% to 65% of all genera detected in those microbiotas in all patients or study subgroups at T_0_ or T_1_). However, some taxa were detected only in the total stool microbiota (from 28% to 43%) or in the Ig-bound stool microbiota fraction (from 3% to 8%). The unique genera in the Ig-bound stool microbiota may be associated with the process of fraction purification that could lead to its enrichment in rare microorganisms, enabling their identification in the Ig-bound stool microbiota, but masked by the diversity in the total stool microbiota (amplification bias, signal dilution).

Spearman’s Rank Correlation Test indicated positive correlations between the relative abundance median values of bacteria (at the genus level) in the Ig-bound and total stool microbiota in all advanced melanoma patients undergoing anti-PD-1 therapy (Figure 8A,B), in the subgroups with favorable clinical outcomes, i.e., in the CB (Figure 8C,D) and R subgroups (Appendix A), and those with unfavorable ones, i.e., the NB (Figure 8E,F) and NR subgroups (Appendix A) at T_0_ and T_1_, respectively (*p*-value ≤ 0.05). It is worth noting that, in the CB subgroup, Spearman’s *ρ* value increased during treatment (Figure 8C,D), while in the NB subgroup, there was an opposite trend (Figure 8E,F). Consistent tendencies were observed in the R and NR subgroups (Appendix A). However, they were less noticeable in the NR than in the NB subgroup (Appendix A).

Furthermore, there were correlations between particular relative abundances of bacteria in the Ig-bound and total stool microbiota (at the amplicon sequence variant (ASV) level) in the R and NR subgroups at T_0_ (*p*-value ≤ 0.05). The Differential Gene Correlation Analysis (DGCA) indicated that 1201 of these correlations differed between those subgroups and 248 correlations, among them, were opposed (*p*-value ≤ 0.05, Appendix A). For instance, there was a positive correlation between the relative abundances of *Christensenellaceae* R-7 group sp. in the total stool microbiota and *Akkermansia muciniphila* in the Ig-bound stool microbiota in the NR subgroup at T_0_, while in the R subgroups at T_0_, the correlation between these bacterial taxa was negative. Similarly, the relative abundance of *Afipia* sp. in the total stool microbiota positively correlated with the relative abundance of *Bifidobacterium bifidum* in the Ig-bound stool microbiota in the NR subgroup, whereas in the R subgroup–negatively. On the other hand, in the R subgroup, the positive correlations were found, e.g., between the relative abundances of *Bacteroides* sp. in the total stool microbiota and *Bifidobacterium* sp. in the Ig-bound stool microbiota or between the relative abundances of *Merdibacter* sp. and *Faecalibacterium prausnitzii* (in the total stool microbiota and Ig-bound stool microbiota, respectively). In the NR subgroups, these correlations were negative. Moreover, there were correlations between the relative abundances of identical ASVs in the Ig-bound and total stool microbiota (*p*-value ≤ 0.05, Appendix A). In detail, in the R subgroup at T_0_, there were strong and positive correlations between the relative abundances of *Bacteroides* sp., *Clostridium sensu* stricto 1 sp., *Christensenellaceae* R-7 group sp., *Enterococcus* sp., and *Bacteroides uniformis* in the total stool microbiota and Ig-bound stool microbiota. In the NR subgroup at T_0_, there was a weak and negative correlation between the relative abundances of *Bacteroides* sp. in the total and Ig-bound stool microbiota. Furthermore, there was medium and positive correlation between the relative abundances of *Clostridium sensu* stricto 1 sp. and *Faecalibacterium prausnitzii*, and strong and positive correlation between the relative abundances of *Erysipelotrichaceae* UCG-003 bacterium and *Parabacteroides distasonis* in the total stool microbiota and Ig-bound stool microbiota. The DGCA showed that the correlations between the relative abundances of all those ASVs differed between the R and NR subgroups at T_0_ (*p*-value ≤ 0.05, Appendix A). It is worth noting that there was an opposite trend in the correlation between relative abundances of *Bacteroides* sp. in the Ig-bound and total stool microbiota between those subgroups. The DGCA results are available in Appendix A.

The comparison of the Ig-bound and total stool microbiota composition corroborated our previous observations that Igs targeted a specific subset of microorganisms, which led to changes in the microbial relative abundance patterns in the Ig-bound stool microbiota vs. total stool microbiota. Interestingly, a decreasing correlation between the relative abundance median values in the Ig-bound and total stool microbiota during anti-PD-1 therapy was observed in patients with PD, while an inverse trend was observed in patients with favorable clinical outcomes. That indicated the differences in the functionality of intestinal immunity between the study subgroups. Moreover, there were correlations between the total stool microbiota and Ig-bound stool microbiota members, which underlined the link between the gut microbiota and intestinal immunity. Their mutual interactions could affect systemic immune responses (or resulted from the alterations in them) and, consequently, the clinical outcomes of the immunotherapy.

## 3. Discussion

Although the association between the gut microbiota composition and clinical outcomes of ICIs was demonstrated in various cohorts of cancer patients [1,2,3,4,5], the link between the intestinal barrier functionality (that mutually interacts with the gut microbiota) and the immunotherapy efficacy has not been extensively studied so far. Therefore, in this study, the intestinal barrier state biomarkers, fecal zonulin, fecal calprotectin, and fecal SIgA, were quantified in advanced melanoma patients receiving anti-PD-1 therapy, before the start and during treatment, to analyze their association with clinical outcomes. As a result, there were found no statistically significant differences in the intestinal barrier permeability and inflammation (reflected by the concentrations of fecal zonulin [10] and calprotectin [12], respectively) between patients with favorable and unfavorable clinical outcomes (Figure 1A–D). However, the level of fecal SIgA at baseline was significantly higher in the R and CB subgroups as compared to the NR and NB subgroups, respectively (Figure 1E,F), suggesting an association between intestinal immunity and response/clinical benefit from the anti-PD-1 therapy in the study cohort. Moreover, the investigated intestinal barrier state biomarkers were significantly associated with survival outcomes. In detail, high baseline concentration of fecal zonulin predicted longer OS, high baseline fecal SIgA level—longer PFS and OS, whereas high baseline concentration of calprotectin—shorter PFS and OS (Figure 3). It can be hypothesized that increased baseline SIgA production could provide homeostatic relationship between the gut microbiota and the host and sustained anti-cancer immune activity. On the other hand, increased permeability may be associated with bacteria and bacterial metabolites translocation from the gut lumen to the lymph nodes and blood. This could induce higher SIgA generation at intestinal mucosa, but also shape the systemic immunity and, consequently, the tumor microenvironment (induction of immune cell differentiation and translocation to the tumor side and activation of anti-tumor immune responses through potential molecular mimicry between commensal bacteria and tumor cells [22]). Moreover, bacterial metabolites in blood plasma, such as short-chain fatty acids, were associated with improved clinical outcomes of ICIs in cancer patients [23]. In contrast, intestinal inflammation could lead to the exhaustion of the immune system and, therefore, might be associated with unfavorable survival outcomes [24]. In line with these findings, other studies also reported some differences in the functioning of the intestinal barrier between cancer patients with favorable and unfavorable clinical outcomes of ICIs. For instance, it was shown that NSCLC patients with baseline plasma citrulline levels higher than 20 μM (reflecting properly functioning enterocytes) more frequently received clinical benefits from the anti-PD-1 therapy and had longer PFS and OS than those with lower citrulline levels [8]. Moreover, in a cohort of melanoma patients receiving the anti-PD-1 therapy, patients with PD had an increased abundance of inflammatory cells in their fecal samples compared to non-progressors [2]. Another study reported lower baseline fecal calprotectin levels in patients with hepatocellular carcinoma (HCC) who clinically benefited from the ICI therapy (anti-CTLA-4 and/or anti-PD-L1 therapy) in comparison to those with PD [24]. Moreover, trends in the changes in fecal calprotectin levels during treatment in patients with HCC were comparable to those observed in the intestinal permeability biomarkers, i.e., serum zonulin and lipopolysaccharide binding protein (LBP), and opposite to those observed in *Akkermansia* to *Enterobacteriaceae* ratio and the gut microbiota alpha diversity. In this study, a positive correlation between fecal zonulin and fecal calprotectin levels was observed in all samples and patients with favorable clinical outcomes at baseline (Figure 2C,H,I). In line with our findings, Coutzac et al. (2020) found that serum zonulin correlated with serum inflammatory proteins, such as tumor necrosis factor-α (TNFα) and monocyte chemoattractant protein 1 (MCP-1) in metastatic melanoma patients undergoing anti-CTLA-4 therapy [25]. In the study cohort, fecal calprotectin also positively correlated with the F/B ratio in the total stool microbiota in all samples and the R subgroup at T_0_ and T_1_ (Figure 2B,F,G). The F/B ratio is considered a biomarker of intestinal homeostasis, and any disruptions in its value may reflect gut microbiota dysbiosis and have been reported in various diseases, such as obesity, inflammatory bowel disease (IBD), and type 2 diabetes (T2D) [19,20]. It is worth noting that the relative abundance of *Bacteroidota* (formerly *Bacteroidetes*, Appendix A) and *Bacteroidetes* to *Firmicutes* (B/F) ratio were increased in the baseline total stool microbiota of the R subgroup compared to the NR subgroup (as reported in our previous paper [3]). These findings imply the link between the total stool microbiota enrichment in *Bacteroidota* phylum members and lower intestinal inflammation and favorable clinical outcomes of anti-PD-1 therapy in the study cohort. In line with that, the association between decreased B/F ratio in the total stool microbiota and intestinal and systemic inflammation (reflected by fecal calprotectin and plasma C-Reactive Protein (CRP) levels, respectively) was indicated in obese subjects [26]. Consistently, several *Bacteroidota* phylum members were found to possess anti-inflammatory and epithelium-reinforcing properties [27]. On the other hand, the F/B ratio in the total stool microbiota was negatively associated with fecal SIgA in all samples and patients with unfavorable clinical outcomes at baseline (Figure 2A,D,E). A similar trend was observed in a study on a porcine model [28]. Moreover, another study demonstrated that the *Bacteroides ovatus*, a member of the *Bacteroidota* phylum, elicited higher fecal SIgA production in mice than other analyzed species [29]. Noteworthy, the concentration of fecal SIgA tended to be lower after monocolonization with *Bacillota*, *Actinomycetota*, or *Pseudomonadota* phylum representatives compared to *Bacteroidota* phylum members. Overall, in this study, there were different trends in the correlation between analyzed biomarkers in patients with favorable and unfavorable clinical outcomes (Figure 2), which implies activation of distinct mechanisms within the intestinal mucosa in those subgroups.

Collectively, these findings imply that the intestinal barrier may affect treatment efficacy in cancer patients receiving ICIs, potentially through the interaction with the gut microbiota and mucosal immunity, and consequently modulate immune responses at the tumor site. Alternatively, trends observed in the functioning of the intestinal barrier in patients with favorable and unfavorable clinical outcomes of anti-PD-1 therapy may reflect the overall immune activation and the ability of the patient’s immune system to fight cancer. Melanoma is known as one of the most immunogenic tumors [30]. It interacts with the host immune system, which allows for cancer cell recognition and attack by immune cells. On the other hand, numerous immunomodulatory mechanisms induced by malignant cells may lead to immune resistance and immunosuppression, favoring tumor progression. Tumor- and host-associated immune signatures may also affect clinical outcomes of the ICI therapy [31]. It is probable that high baseline levels of fecal SIgA and zonulin may reflect overall melanoma-induced activation of the immune system, which may favor efficacy of the immunotherapy, while low levels of these biomarkers at baseline may mirror immune exhaustion or immunosuppression that may negatively impact clinical outcomes of ICIs. Similarly, a high baseline level of fecal calprotectin may reflect impaired anti-cancer responsiveness of the immune system. To have a better insight into the link between intestinal barrier state and systemic immunity at baseline in the study cohort, the association between baseline peripheral blood cell counts and ratios with clinical outcomes and intestinal barrier state biomarker levels was analyzed. Additionally, the mutual correlations between blood and fecal indicators were investigated. There was significantly higher median absolute basophil count (ABC) in patients with high (>303.4 ng·mL^−1^) fecal zonulin levels at baseline as compared to those with lower zonulin levels (*p*-value = 0.01, Appendix A). Moreover, intestinal barrier state biomarkers moderately correlated with peripheral blood cell counts and ratios (*p*-value ≤ 0.05, Appendix A), which levels, according to literature, reflect enhanced or suppressed anti-tumor immune responses and are associated with clinical outcomes of ICIs [32,33,34,35,36,37]. However, the correlations found in patients with favorable clinical outcomes were distinct from those observed in the subgroups with unfavorable ones (Appendix A). Considering this, the potential of intestinal barrier state biomarkers as predictive/prognostic biomarkers in cancer patients undergoing ICI therapy appears to be significant and warrants further investigation. Moreover, several aspects should be considered during the evaluation of biomarker clinical utility and the establishment of accurate cut-off points for patient stratification. Firstly, in the current study, the biomarker concentrations were elevated in most patients, and estimated biomarker cut-off levels that predicted improved or poor survival outcomes were also increased (from 2.7 to 4.2 times higher than upper limit of the normal range), regarding the reference range [9,16,17], apart from the concentration of fecal SIgA predicting longer PFS, which was within normal limits [16]. These results revealed disrupted intestinal barrier functionality in advanced cancer patients, implying that specific reference ranges should be established for such cohorts. Secondly, elevated concentrations of investigated biomarkers may not only be associated with tumors, but also with coexisting diseases, such as autoimmune, infective, or metabolic diseases [10,12,38]. Thirdly, other factors may change the concentration of intestinal barrier state biomarkers, such as age [9,39], tumor stage [38], and diet [40,41]. On the other hand, it should be investigated whether the intestinal barrier and immunity may become a therapeutic targets to improve clinical outcomes of ICIs in cancer patients. There are several potential interventions to modulate their functioning, e.g., through pharmaceutical intervention, prebiotics and probiotics, diet and supplements, and fecal microbiota transplantation (FMT) [42]. For instance, Renga et al. (2022), in a study on a murine model of melanoma, demonstrated that indole-3-carboxaldehyde (known to contribute to the maintenance of the intestinal barrier homeostasis) prevented intestinal damage associated with ICI-induced colitis and simultaneously did not interfere with the anti-cancer activity of anti-CTLA-4 therapy [43].

As baseline concentration of fecal SIgA was increased in patients with favorable clinical outcomes (Figure 1E,F) and its high baseline level correlated with longer PFS and OS (Figure 3I–L) in the study cohort, the Ig-bound fraction of total stool microbiota was purified, and its bacterial composition alongside the total stool microbiota composition was analyzed. The Ig-bound stool microbiota fraction could consist of microorganisms coated with IgA, IgG, and IgM. However, those coated with IgAs should constitute the largest fraction as SIgA is the dominant Ig produced in the intestinal secretions, while both IgG and IgM were found to be produced in much lower amounts in the gut [13].

The analysis indicated that the Ig-bound stool microbiota of advanced melanoma patients undergoing anti-PD-1 therapy was dominated by *Bacillota* phylum members regardless of clinical outcome or collection time point (Figure 4B,D). For comparison, the *Bacillota* phylum was also the dominant one in the total stool microbiota; however, its dominance was less evident (Figure 4A,C). Noteworthy, the *Bacillota* phylum that elicited Ig responses in the study cohort, was enriched in the baseline fecal microbiota of metastatic melanoma patients, who benefited from the anti-CTLA-4 therapy and more frequently experienced treatment-induced colitis [44]. These findings demonstrated that *Bacillota* phylum members were associated with intestinal and systemic immunity. Consistently with the trends observed at the phylum level, beta diversity analysis indicated that the Ig-bound stool microbiota and the total stool microbiota shared a substantial subset of bacterial genera (Figure 7A); however, the genera’ relative abundance patterns differed between them (Figure 7B). In line with that, some bacterial genera were detected only in the total or Ig-bound stool microbiota. While the presence of unique taxa in the total stool microbiota implies that a subset of bacteria was not coated with Igs, the detection of unique genera in the Ig-bound stool microbiota was unexpected. Differences in the genera’s relative abundance patterns between the microbiotas may resulted from the variations in the induction of Ig responses between bacterial genera, as it was demonstrated by Yang et al. (2020) [29]. On the other hand, they (and the detection of unique genera in the Ig-bound stool microbiota) may be associated with the process of fraction purification that could lead to the enrichment in rare microorganisms. As a result, efficient amplification of their sequences and identification were feasible in the Ig-bound stool microbiota, but masked by the diversity in the total stool microbiota. It was shown that primers used for amplification may preferentially bind to certain sequences, leading to differential amplification efficiency. Consequently, some microbial groups could be overrepresented while others, particularly rare taxa, underrepresented or missed entirely [45]. Furthermore, factors such as the number of PCR cycles, annealing temperatures, and the presence of inhibitors could introduce biases during amplification. Overamplification could lead to the dominance of certain sequences, masking the presence of rare microorganisms [46].

Furthermore, it was found that the relative abundance median values of bacterial genera in the Ig-bound and total stool microbiota correlated positively in all advanced melanoma patients undergoing anti-PD-1 therapy and subgroups of patients with favorable and unfavorable clinical outcomes at T_0_ and T_1_ (Figure 8 and Appendix A). However, Spearman’s *ρ* value have changed during treatment, and trends in the changes were opposite between patients with distinct clinical outcomes (Figure 8 and Appendix A). These findings suggest that there is an association between the anti-PD-1 therapy and Ig production in the intestines. Consistently, changes in the Ig-bound stool microbiota composition were observed during the anti-PD-1 therapy at the phylum (Figure 5) and genus level (the DAA, Figure 6). These changes were more noticeable in patients with unfavorable clinical outcomes than those with favorable ones (Figure 5 and Figure 6). The DAA also revealed that the highest variance in the relative abundance of bacterial genera in the Ig-bound stool microbiota fraction between patients with favorable and unfavorable clinical outcomes was at baseline (expressed as the number of differentially abundant genera); during treatment, the differences in the microbial signatures between subgroups were less evident (Figure 6). Noteworthy, fecal SIgA level was also higher at baseline in patients with favorable clinical outcomes as compared to those with unfavorable ones. These findings also imply that anti-PD-1 therapy could exert an impact on the Ig response (directly affecting the intestinal immunity or through the modification of general immunological status of patients). Previous studies demonstrated the association between PD-1 expression on Peyer’s patches (PPs) T cells and IgA production [47,48,49]. Kawamoto et al. (2012) indicated that PD-1 deficiency was associated with an excessive generation of T follicular helper (T_FH_) cells with altered phenotypes that lead to impaired selection of IgA^+^ B cells in the germinal centers (GCs) of PPs [47,48]. As a result, the IgAs produced in PD-1-deficient mice had reduced bacteria-binding capacity. That caused compositional alterations in the gut microbiota (a decrease in the anaerobic bacteria, *Bifidobacterium* genus, and *Bacteroides* genus, and an increase in the *Enterobacteriaceae* family as compared to wild-type mice). Gut microbiota dysbiosis and associated disruption of epithelial integrity resulted in an excessive activation of the whole-body immune system and an expansion of self-reactive B and T cells and auto-antibody production. Additionally, Zhang et al. (2015) demonstrated that intestinal ischemia/reperfusion in mice decreased PD-1/PD-L1 interaction on PP CD4^+^ T cells [49]. Lower PD-1/PD-L1 expression correlated with reduced production of cytokines involved in the proliferation and differentiation of IgA^+^ B cells, such as transforming growth factor-β1 (TGF-β1) and interleukin-21 (IL-21), impaired intestinal IgA synthesis, and mucosal integrity. Collectively, these findings suggest that PD-1/PD-L signaling plays an important role in the regulation of IgA production and, consequently, affects the composition of the gut microbiota, intestinal barrier, and systemic immunity. All of these factors were found to exert an impact on clinical outcomes of ICIs in cancer patients [8,31].

To our knowledge, the influence of anti-PD-1 therapy on SIgA production in the intestinal mucosa and, consequently, its effect on treatment efficacy has not been studied so far. Therefore, observed associations allow us to propose several new hypotheses that require verification. Changes in the Ig-bound stool microbiota composition during treatment (Figure 5, Figure 6, Figure 8 and Appendix A), especially those observed in patients, who did not respond or benefit from the anti-PD-1 therapy, indicate the existence of a possible mechanistic link between the action of the therapeutic anti-PD-1 antibodies and SIgA production. That could affect the gut microbiota composition and/or activity, and consequently influence the whole-body immune system and clinical outcomes of anti-PD-1 therapy. In line with that, the composition of the total stool microbiota has changed during treatment, i.e., there were alterations in the relative abundance patterns at the phylum level (Appendix A) and at the genus level indicated in the DAA (Appendix A). Noteworthy, changes in the total stool microbiota composition during the anti-PD-1 therapy, similarly to trends observed in the Ig-bound stool microbiota fraction, were more evident in patients with unfavorable clinical outcomes (Appendix A), supporting our assumption that anti-PD-1 antibodies could affect the gut microbiota through the disruption of IgA generation. On the other hand, no statistically significant changes were found in the intestinal barrier state biomarker levels during treatment within study subgroups (Figure 1), which indicates a lack of profound influence of the anti-PD-1 therapy on the intestinal barrier functionality. However, it is worth noting that the median fecal SIgA concentration tended to decrease at T_n_ (as compared to the baseline median level) in the R subgroup, while in the NR subgroup, it increased ~2 times at T_1_, being comparable to that observed in the R subgroup at T_1_, and decreased at T_n_, being comparable to that observed at baseline (Figure 1). Accordingly, Kawamoto et al. (2012) reported that despite reduced bacteria-binding capacity in the PD-1-deficient mice, the concentration of free IgA in intestinal secretions was higher in them than in wild-type mice [47,48].

Furthermore, the alterations in the Ig responses could also affect the functional potential of the gut microbiota in advanced melanoma patients, and, therefore, influence the systemic immunity and clinical outcomes of the immunotherapy. It was reported that bacterial antigen coating by SIgA may affect the function of the targeted microbe [50], and not necessarily its abundance. Briefly, Rollenske et al. [50], in a study on a murine model monocolonized with *Escherichia coli*, demonstrated that exposure of the intestinal mucosa with a single transitory microbe induced the generation of antigen-specific dimeric SIgA that targeted a wide range of membrane-associated antigens. Monoclonal IgA (mIgA) binding to bacterial antigens exerted distinct alterations in microbial function and metabolism (also when mIgAs targeted the same antigens). The outcome on the target bacterium was shaped by the context and specificity of mIgAs. On the other hand, surface-binding mIgAs induced generic functional effects on the bacteria associated with their motility and susceptibility to bile acids. In this study, we found numerous correlations between the particular relative abundances of bacterial ASVs in the Ig-bound and total stool microbiota in the R and NR subgroups at baseline (Appendix A). Moreover, these correlations differed between those subgroups, and a subset of them was opposed. Among bacterial taxa in the Ig-bound stool microbiota that correlated with taxa in the total stool microbiota, and their correlations were opposed in the R vs. NR subgroups, there were those indicated in previous studies as favorable or unfavorable in cancer patients undergoing ICI therapy, such as *Akkermansia muciniphila*, *Faecalibacterium prausnitzii*, *Bacteroides* sp., and *Bifidobacterium* sp. [51,52,53,54,55]. Considering the findings of Rollenske et al. (2021) [50], observed correlations indicate that the presence or absence of bacterial taxa does not fully reflect their functional activities and their influence on anti-cancer immune responses (both favorable and unfavorable), as they may be changed by Ig coating. Despite existing differences in the structure and physiology of the gastrointestinal and immune system between rodents and humans, it can be expected that the mechanisms regulating the interaction between the gut microbiota and the host that developed during the co-evolution are similar. To our knowledge, this study is the first one that investigated the composition of the Ig-bound stool microbiota fraction in advanced melanoma patients undergoing anti-PD-1 therapy and shed light on the association between intestinal immunity and clinical outcomes of the immunotherapy. Future studies are required to provide mechanistic insight into the association between PD-1 antibodies and SIgA production and the relationship between intestinal immunity, gut microbiota, and anti-cancer immune responses. Moreover, it should also be investigated whether the introduction of SIgAs targeting specific microbial antigens (that may induce diverse functional changes [50]) could serve as a strategy to improve the clinical outcomes of the ICI therapy. Richards et al. (2021) demonstrated that passive oral administration of SIgA has the potential to combat pathogenic species [56]. If the targeted modification of the gut microbiota through SIgAs was achievable, it would provide an alternative to conventional FMT, which possess several limitations, such as risk of pathogen transfer, stool toxicity, low reproducibility, and transient long-term outcome [57].

This study has several limitations. Firstly, the study comprised a small number of patients, which limited the power of statistical analyses. Secondly, the V3–V4 region of 16S rRNA gene sequencing, which was performed to analyze the composition of the Ig-bound and total stool microbiota, allowed only for the detection of bacterial and archaeal sequences, while viruses and fungi could also be present in these microbial communities. Therefore, it is recommended to use whole-genome sequencing to provide a comprehensive analysis of the composition and functional potential of the investigated microbiotas. Thirdly, the specificity of Igs, their microbe-binding capacity, and antigens that elicited Ig responses (potential food allergies) were not analyzed in this study.

## 4. Materials and Methods

### 4.1. Study Cohort Description and Collection of Clinical Data and Stool Samples

The study cohort comprised 64 patients with histologically confirmed unresectable stage III or stage IV cutaneous melanoma enrolled in treatment with anti-PD-1 therapy, i.e., nivolumab or pembrolizumab (as a part of the Ministry of Health (Poland) drug program [58]). Recruitment to the study was conducted at the Department of Medical and Experimental Oncology, Heliodor Święcicki Clinical Hospital, Poznań University of Medical Sciences (Poznań, Poland) between June 2018 and December 2021. Written informed consent was obtained from all participants. The study was approved by the Bioethics Committee at Poznań University of Medical Sciences (registration number 402/18).

Clinical information, including tumor stage, baseline serum LDH concentration, baseline peripheral blood cell counts, response, PFS, and OS, was collected from the medical records. Response to the anti-PD-1 therapy was assessed according to the response evaluation criteria in solid tumors (RECIST) v.1.1. Patients were classified as R (complete response (CR) and partial response (PR)) or NR (disease stabilization (SD) and PD). Additional classification categorized patients into CB (CR, PR, and SD) or NB (PD). Outcome descriptors are presented in the Appendix A. Patients’ categorization into the subgroups at baseline was performed concerning the best overall response they have experienced during treatment. Survival analysis was based on two clinical endpoints, i.e., PFS and OS. PFS was defined as the time from the anti-PD-1 therapy initiation to the first event, i.e., disease progression or death from any cause. OS was defined as the duration of patient survival from initial anti-PD-1 therapy. Baseline characteristics of the study cohort are presented in our previous paper [3].

Stool samples were collected from patients before the anti-PD-1 therapy initiation (T_0_, at baseline) and during treatment (T_1_ and T_n_, approximately 3 and later than 3 months, mostly about 12 months, from the start of the anti-PD-1 therapy, respectively). The exact procedure of fecal sample collection was described in our previous paper [3]. Briefly, study participants were requested to collect feces according to the stool collection procedure described by Dore et al. (2015) [59] and deliver the samples to the hospital at scheduled time points. Then, the samples were transported to the laboratory, where they were portioned and stored for further analysis.

### 4.2. Measurement of the Intestinal Barrier State Biomarker Concentration

The concentration of fecal SIgA and fecal calprotectin was measured with commercially available ELISA kits, i.e., IDK^®^ SIgA ELISA Kit (cat. no. K8870) and IDK^®^ Calprotectin ELISA Kit (cat. no. K6927, Immundiagnostik AG, Bensheim, Germany), respectively. Fecal zonulin level was measured with a competitive ELISA kit—IDK^®^ Zonulin ELISA (cat. no. K5600, Immundiagnostik AG, Bensheim, Germany). All measurements were performed according to the manufacturer’s protocols.

### 4.3. Purification of Ig-Bound Stool Microbiota Fraction

The procedure of purification of Ig-bound stool microbiota fraction from the total stool microbiota was based on the methodology described by Madhwani et al. (2016) [60]. Briefly, biotinylated antibodies specific to human Igs (goat anti-human IgG, IgM, IgA (H+L), Biotin, cat. no. 31782, Invitrogen, Waltham, MA, USA) were incubated with streptavidin-coated magnetic beads (Streptavidin Magnetic Particles, cat. no. 116417786001, Roche, Basel, Switzerland). Then, excessive or non-specifically bound antibodies were removed by washing the beads three times with the buffer (20 mM potassium phosphate, 0.15 M sodium chloride, pH 7.5). In the next step, the antibodies coupled with the magnetic beads were incubated with the fecal suspension in the same buffer to separate the stool microbiota fraction coated with Igs from the total stool microbiota. Then, the Ig-bound stool microbiota fraction coupled with the anti-human Igs—magnetic bead complexes were washed three times in the buffer with 0.1% bovine serum albumin and 0.05% Tween 20 to remove non-specifically bound bacteria. These steps were performed using a magnetic separator, MagMAX™ Express (Applied Biosystems, Waltham, MA, USA). After the process, metagenomic DNA was isolated from the purified fraction of Ig-bound stool microbiota using the commercial system DNeasy PowerSoil Pro Kit (cat. no. 47016, Qiagen, Hilden, Germany), following the manufacturer’s protocol.

### 4.4. Metagenomic DNA Extraction and Sequencing of the V3–V4 Regions in the 16S rRNA Gene

Simultaneously, metagenomic DNA was also extracted from ~250 mg of RNAlater-preserved fecal slurry (total stool microbiota), using DNeasy PowerSoil Pro Kit according to the kit manual (with an additional step of RNase digestion after the bead-beating; detailed procedure was described in our previous paper [3]). The concentration of metagenomic DNA extracted from total stool and Ig-bound stool microbiota was determined fluorometrically using the QuantiFluor dsDNA System (cat. no. E2670, Promega, Madison, WI, USA), following the manufacturer’s instructions. Then, all DNA templates were diluted to a concentration of ~5 ng × µL^−1^ before sequencing.

The detailed procedure of library preparation and marker gene sequencing was described in our previous paper [3]. Briefly, the V3–V4 hypervariable regions of the 16S rRNA gene were amplified using the 341F and 785R primers [61] and KAPA HiFi HotStart Ready Mix (cat. no. 07958927001, Roche, Basel, Switzerland) according to the manufacturer’s instructions. Then, the PCR amplicons were delivered to Genomed S.A. (Warsaw, Poland), where the second stage of library preparation and metagenome sequencing on the Illumina MiSeq PE300 platform was performed.

Subsequently, the sequences of V3–V4 regions in the 16S rRNA gene were processed as reported in our previous paper [3]. However, the taxonomy assigned to each merged sequence was performed using the SILVA SSU database release 138.1 with an emended description of the genus *Lactobacillus* Beijerinck 1901 [62].

### 4.5. Statistical Analysis

The Wilcoxon Rank Sum Test was performed to analyze the differences in the intestinal barrier state biomarker levels between study subgroups; the *p*-value ≤ 0.05 was regarded as significant. The mutual correlations between intestinal barrier state biomarkers and their correlations with the F/B ratio in the total stool microbiota were examined with Spearman’s Rank Correlation Test; the *p*-value ≤ 0.05 was regarded as significant. The F/B ratio was calculated by dividing the relative abundance of *Bacillota* (formerly *Firmicutes*) by the relative abundance of the *Bacteroidota* (formerly *Bacteroides*) in the total stool microbiota [63]. Kaplan–Meier curves of the probability of PFS and OS according to high and low baseline levels of intestinal barrier state biomarkers were compared using the Log-Rank (Mantel–Cox) Test; *p*-value ≤ 0.05 was regarded as significant. The maximally selected rank statistics were used to determine optimal cut-off points of biomarker levels. The Cox proportional hazards regression analysis was performed to examine the effects of high vs. low baseline concentrations of biomarkers on HR (HR > 1 indicates an increased risk of disease progression or death, while HR < 1—a decreased risk); the *p*-value ≤ 0.05 was regarded as significant.

Student’s t-test was performed to analyze the differences in the composition of Ig-bound stool microbiota and total stool microbiota at the phylum level; the *p*-value ≤ 0.05 was regarded as significant. The DAA was performed with ANOVA-like Differential Expression version 2 (ALDEx2) tool to determine differentially abundant taxa at the genus level in the Ig-bound stool microbiota and total stool microbiota between study subgroups. Statistics were performed with the Wilcoxon Rank Sum Test; the *p*-value < 0.1 was regarded as significant. PCoA based on the weighted-UniFrac distance metric and the unweighted-UniFrac distance metric [64] was performed to analyze beta diversity between the Ig-bound stool microbiota, total stool microbiota, and circulating cfmDNA (published in our previous paper [21]) extracted from blood plasma samples of the study cohort. Spearman’s Rank Correlation Test was performed to analyze the correlations between relative abundance median values of genera in the Ig-bound stool microbiota and total stool microbiota; the *p*-value ≤ 0.05 was regarded as significant. Spearman’s Rank Correlation Test was also performed to analyze the correlations between particular relative abundances of microbial taxa in the Ig-bound stool microbiota and total stool microbiota; the *p*-value ≤ 0.05 was regarded as significant. The differential correlations between study subgroups were determined with DGCA [65]; the *p*-value ≤ 0.05 was regarded as significant. The data were transformed with the proportion-based ‘The Hellinger’ method [66,67].

All statistical analyses of the data were conducted using R version 4.4.2, and the following R packages were used: phyloseq v.1.50.0, microbiome v.1.28.0, survival v.3.8.3, survminer v.0.5.0, ALDEx2 v1.38.0, and DGCA v1.0.3. 

## 5. Conclusions

The association of intestinal barrier state biomarkers and Ig-bound stool microbiota signatures with clinical outcomes of anti-PD-1 therapy in the study cohort suggests that intestinal barrier functionality may contribute to shaping anti-cancer immune responses in advanced melanoma patients undergoing anti-PD-1 therapy, potentially through the interaction with the gut microbiota and/or whole-body immunity. Alternatively, intestinal barrier state may reflect the overall immune activation and efficacy of patients’ immunity. Future studies that will provide mechanistic insight into this aspect and evaluate the clinical value and utility of indicators of intestinal barrier state as potential biomarkers and/or therapeutic targets are warranted.

## Figures and Tables

**Figure 1 ijms-26-08063-f001:**
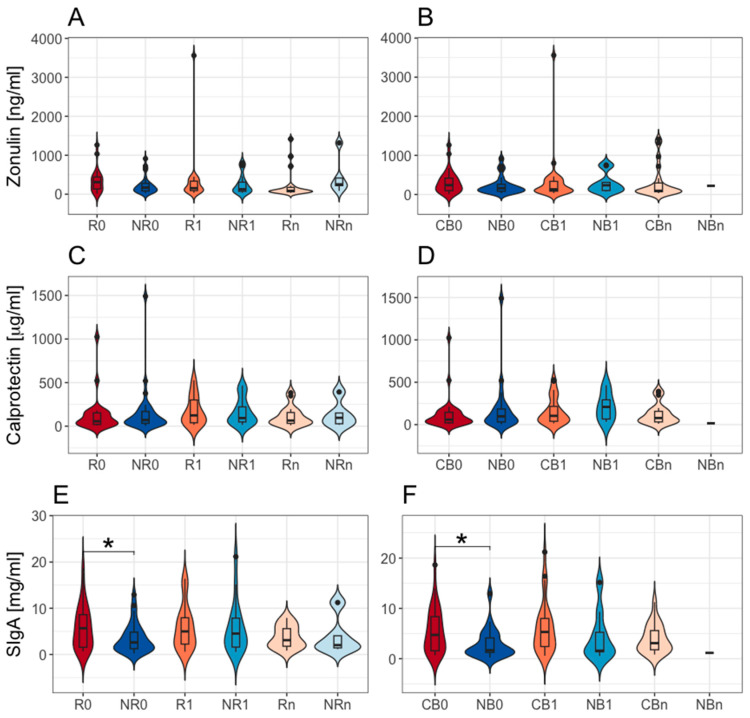
The comparison of intestinal barrier state biomarker concentrations, i.e., fecal zonulin (**A**,**B**), fecal calprotectin (**C**,**D**), and fecal secretory immunoglobulin A–SIgA (**E**,**F**) in advanced melanoma patients before anti-PD-1 therapy initiation—at T_0_ (0) and during treatment—at T_1_ and T_n_ (1 and n, respectively). Patients were classified as responders (R) or non-responders (NR), and patients with clinical benefit (CB) or patients with no clinical benefit (NB), according to the clinical outcome. Statistics were performed using the Wilcoxon Rank Sum Test; *p*-value ≤ 0.05 was regarded as significant (*: *p*-values ≤ 0.05). The figures show that a high baseline level of fecal SIgA was associated with response and clinical benefit from the anti-PD-1 therapy in advanced melanoma patients. Statistically significant differences in the biomarker levels between noncorresponding subgroups, i.e., subgroups with distinct clinical outcomes to the anti-PD-1 therapy at different collection time points, are not marked in the figures. PD-1—programmed cell death protein 1.

**Figure 2 ijms-26-08063-f002:**
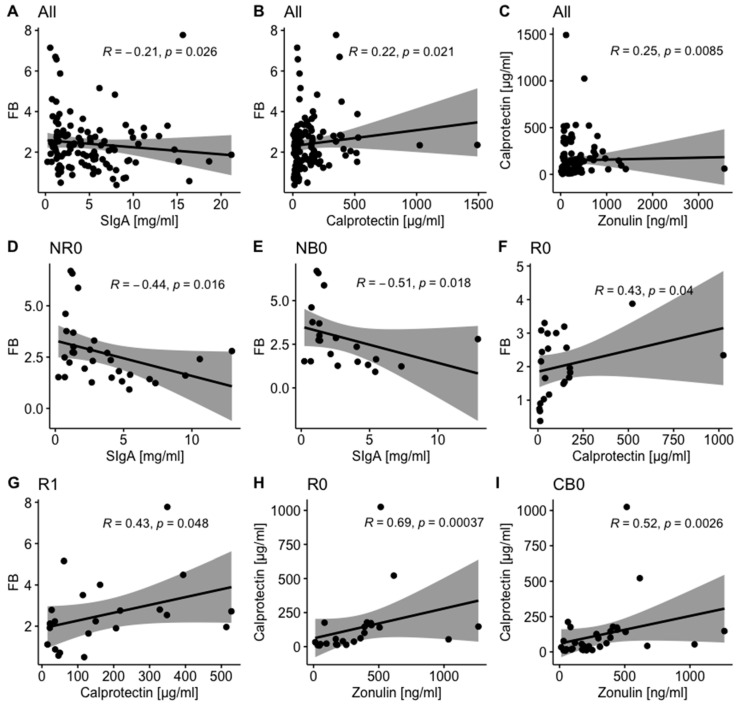
The mutual correlations between intestinal barrier state biomarkers, i.e., fecal zonulin, fecal calprotectin, and fecal secretory immunoglobulin A (SIgA) and *Firmicutes* to *Bacteroidota* (FB) ratio in the total stool microbiota in advanced melanoma patients. The correlations were examined in all analyzed samples and the study subgroups at particular collection time points, i.e., before anti-PD-1 initiation at T_0_ (0) and during therapy at T_1_ and T_n_ (1 and n, respectively). Patients were categorized into subgroups: responders (R) or non-responders (NR) and patients with clinical benefit (CB) or those with no clinical benefit (NB) according to the clinical outcome of the immunotherapy. Statistics were performed using Spearman’s Rank Correlation Test. The *p*-value ≤ 0.05 was regarded as statistically significant. (**A**–**C**) show the trends observed in all analyzed samples; (**D**)—in the NR0 subgroup; (**E**)—in the NB0 subgroup; (**F**)—in the R0 subgroup; (**G**)—in the R1 subgroup; (**H**)—in the R0 subgroup; (**I**)—in the CB0 subgroup. There were found distinct correlations between analyzed biomarkers in advanced melanoma patients with favorable vs. unfavorable clinical outcomes of anti-PD-1 therapy. PD-1—programmed cell death protein 1.

**Figure 3 ijms-26-08063-f003:**
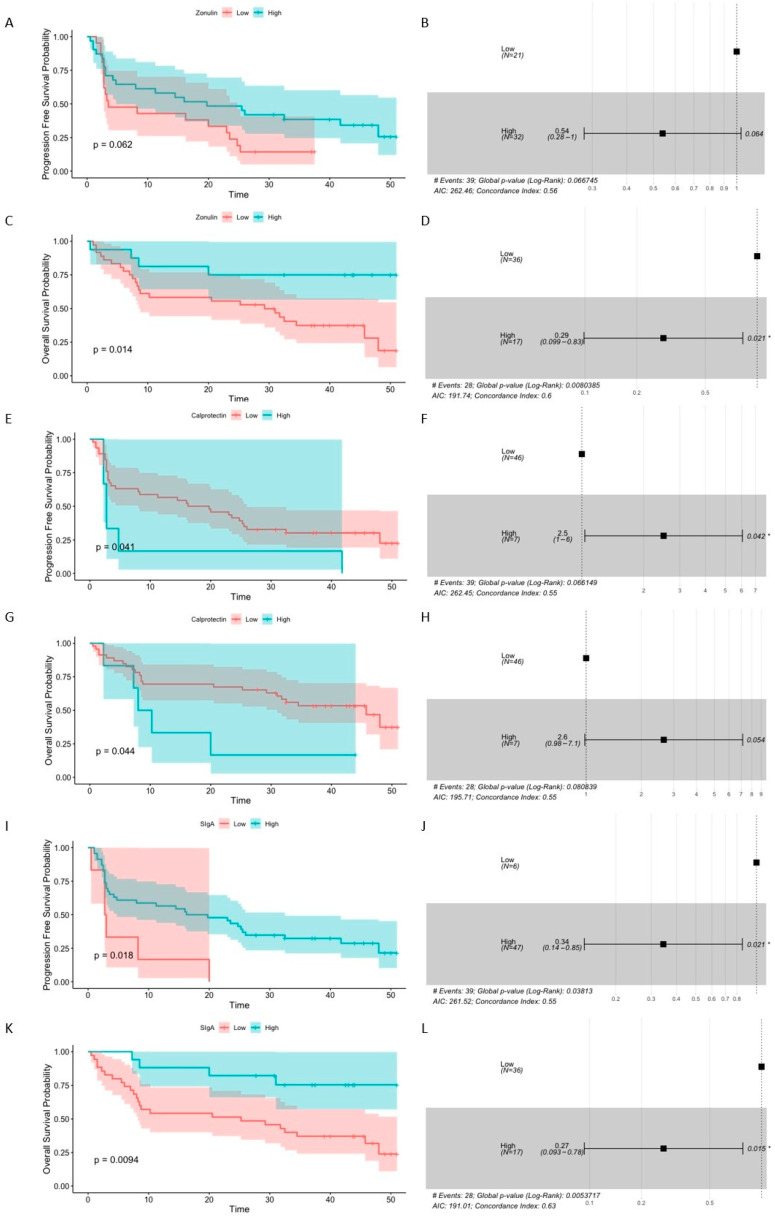
The association between the baseline concentrations of investigated intestinal barrier state biomarkers, i.e., fecal zonulin (**A**–**D**), fecal calprotectin (**E**–**H**), and fecal secretory immunoglobulin A–SIgA (**I**–**L**) and survival outcomes, expressed as progression-free survival (PFS) and overall survival (OS), in advanced melanoma patients enrolled in the treatment with anti-PD-1 antibodies. On the left side of the graph, there are presented Kaplan–Meier curves of the probability of PFS (**A**,**E**,**I**) and OS (**C**,**G**,**K**) according to high and low levels of investigated biomarkers, which were compared using the Log-Rank (Mantel–Cox) Test; *p*-value ≤ 0.05 was regarded as statistically significant. Vertical ticks show censored data. The central line is the median PFS or OS probability, and the shaded area shows a 95% confidence interval. The maximally selected rank statistics were used to determine optimal cut-off points of biomarker levels. The Cox proportional hazard regression was used to examine the effects of high vs. low baseline concentrations of biomarkers on survival outcomes. On the right side (**B**,**D**,**F**,**H**,**J**,**L**), there are presented hazard ratio (HR) and score (log-rank) test two-tailed *p*-value from Cox proportional hazards regression analysis (*: *p*-value ≤ 0.05). HR > 1 indicates an increased risk of disease progression or death, while HR < 1 indicates a decreased risk. The figures show that high fecal zonulin and fecal SIgA levels at baseline were associated with improved survival outcomes in advanced melanoma patients undergoing anti-PD-1 therapy, while high baseline fecal calprotectin level were associated with poor survival outcomes. PD-1—programmed cell death protein 1. # Events—the number of disease progression (for PFS) or deaths (for OS) that occurred within the study cohort during the observation period. AIC—Akaike Information Criterion.

**Figure 4 ijms-26-08063-f004:**
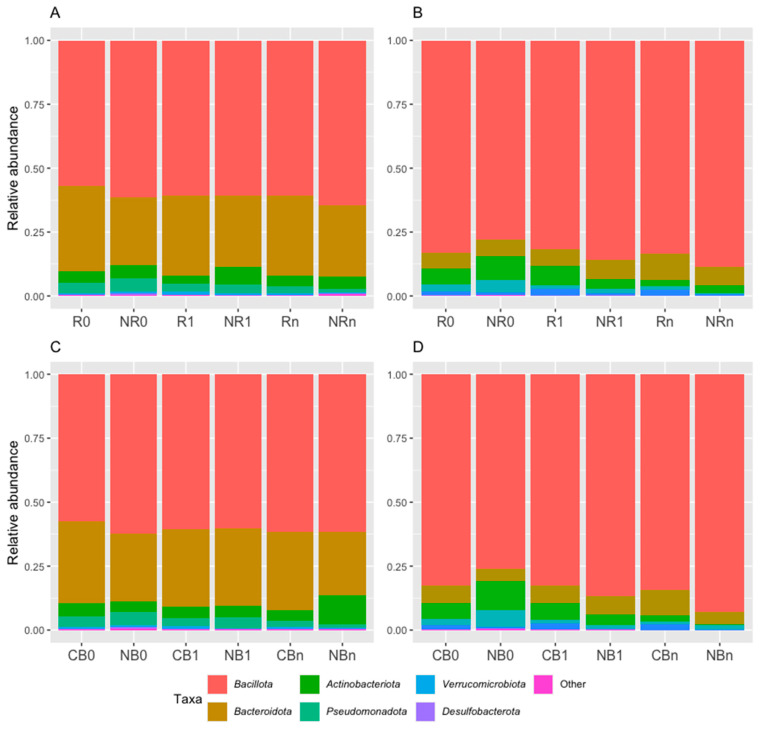
Taxonomic profiles as averaged relative abundances of bacterial taxa in the total stool microbiota (**A**,**C**) and immunoglobulin (Ig)-bound stool microbiota (**B**,**D**) at the phylum level in advanced melanoma patients undergoing the anti-PD-1 therapy, before its start—at T_0_ (0) and during treatment—at T_1_ and T_n_ (1 and n, respectively). Patients were classified as responders—R or non-responders—NR (**A**,**B**) as patients with clinical benefit—CB or patients with no clinical benefit—NB (**C**,**D**) according to the clinical outcome of the immunotherapy. The total stool microbiota was dominated by phyla *Bacillota* (formerly *Firmicutes*) and *Bacteroidota* (formerly *Bacteroidetes*), while in the Ig-bound stool microbiota, there was a remarkable dominance of phylum *Bacillota* members in all study subgroups. PD-1—programmed cell death protein 1.

**Figure 5 ijms-26-08063-f005:**
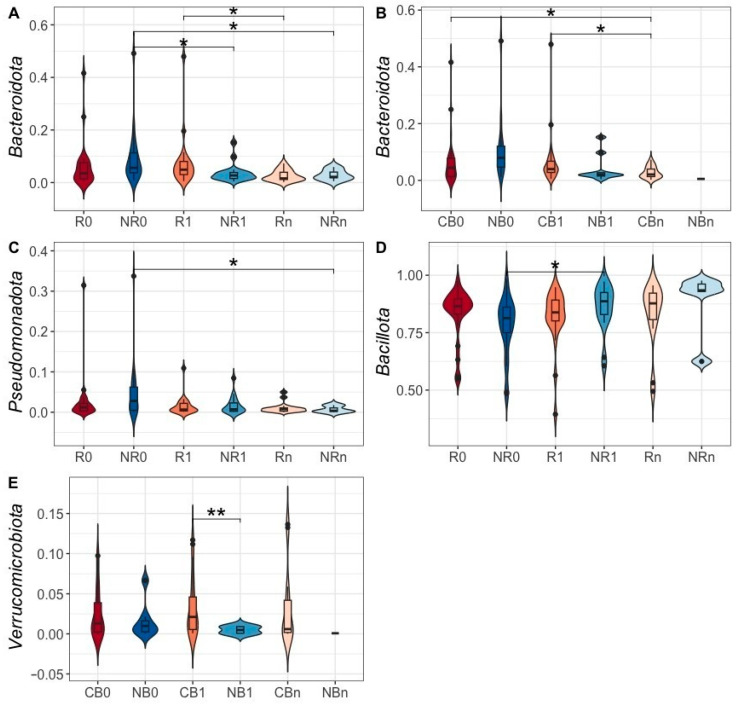
The comparison of the relative abundances of phyla: *Bacteroidota*—formerly *Bacteroidetes* (**A**,**B**), *Pseudomonadota*—formerly *Proteobacteria* (**C**), *Bacillota*—formerly *Firmicutes* (**D**), and *Verrucomicrobiota*—formerly *Verrucomicrobia* (**E**) in the immunoglobulin (Ig)-bound stool microbiota between advanced melanoma patients receiving the anti-PD-1 therapy, before the its start at T_0_ (0) and during treatment at T_1_ and T_n_ (1 and n, respectively). Patients were classified as responders—R or non-responders—NR (**A**,**C**,**D**) and patients with clinical benefit—CB or patients with no clinical benefit—NB (**B**,**E**) according to the clinical outcome of the immunotherapy. The *p*-values describing the statistical significance of the differences in the relative abundances of particular phyla between study subgroups were calculated with Student’s t-test. The *p*-value ≤ 0.05 was regarded as significant (*: *p*-value ≤ 0.05, **: *p*-value ≤ 0.01). Statistics indicated a decrease in the relative abundances of phyla Bacteroidota and Pseudomonadota and an increase in phylum Bacillota during anti-PD-1 therapy in the NR subgroup. A decrease in the relative abundance of Bacteroidota was also observed in the R and CB subgroups during treatment. Moreover, there was a higher relative abundance of Verrucomicrobiota phylum in the CB vs. NB subgroup at T_1_. PD-1—programmed cell death protein 1.

**Figure 6 ijms-26-08063-f006:**
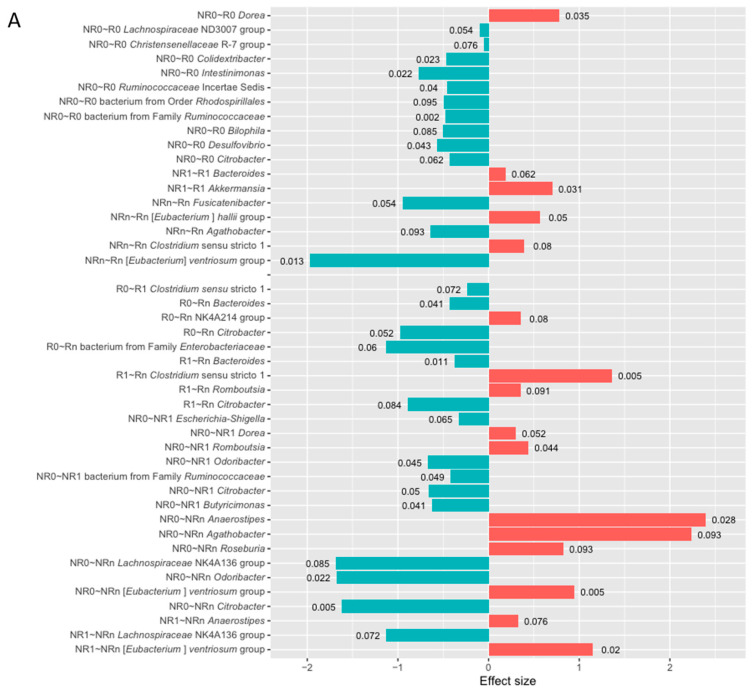
The differentially abundant genera in the immunoglobulin (Ig)-bound stool microbiota between advanced melanoma patients with favorable vs. unfavorable clinical outcomes of the anti-PD-1 therapy, before its start—at T_0_ (0) and during treatment—at T_1_ and T_n_ (1 and n, respectively) identified in the differential abundance analysis (DAA) performed using ANOVA-like Differential Expression version 2 (ALDEx2) tool. Moreover, changes in the relative abundances of genera during treatment (T_0_ vs. T_1_, T_0_ vs. T_n_, and T_1_ vs. T_n_) within those subgroups were also indicated with the DAA. Patients were classified as responders—R or non-responders—NR (**A**) and as patients with clinical benefit—CB or patients with no clinical benefit—NB (**B**) according to the clinical outcome of the immunotherapy. The figures illustrate only the statistically significant results (Wilcoxon Rank Sum Test, *p*-value < 0.1 was regarded as significant). The *p*-values describing the statistical significance of the DAA results were placed at the tips of the bars. The direction of changes in the relative abundances of taxa between the two subgroups being compared was assessed based on the effect size values. The first group of the two being compared is considered a reference group, whereas the second one is a tested group (group designations are placed on the left side of the graph at the beginning of the following lines; ‘reference group~tested group’). A positive effect size (red bars) suggests a higher relative abundance of a particular taxon in the tested group compared to the reference group, while negative (blue bars)—lower. Effect size also measures the biological significance of the observed differences (the larger the effect size, the more substantial the difference between subgroups). The names of the differentially abundant taxa are placed on the left side of the graph (next to the subgroup designation). The DAA indicated that the Ig-bound stool microbiota signatures were associated with clinical outcomes of the anti-PD-1 therapy and have changed during treatment. PD-1—programmed cell death protein 1.

**Figure 7 ijms-26-08063-f007:**
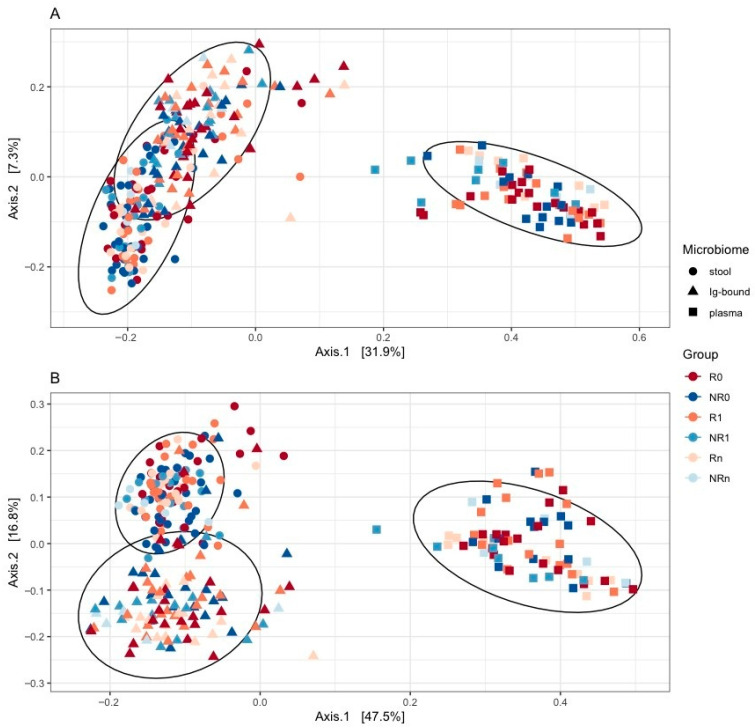
Principal coordinates analysis (PCoA) based on a quantitative distance measures ((**A**); weighted-UniFrac) and a qualitative distance measures ((**B**); unweighted-UniFrac) was performed to compare the similarities and dissimilarities between the immunoglobulin (Ig)-bound stool microbiota (indicated by triangles) and total stool microbiota (indicated by dots) at the genus level (beta diversity analysis). The circulating cell-free microbial DNA—cfmDNA (plasma; indicated by squares) of the same patients (published in a separate paper [21]) was added to the plot for reference purposes to better visualize distances between the microbiotas. The microbiota composition was analyzed in advanced melanoma patients before—at T_0_ (0) and during anti-PD-1 therapy—at T_1_ and T_n_ (1 and n, respectively). Patients were classified as responders (R) or non-responders (NR) according to the clinical outcome of the immunotherapy. The figures demonstrate that the Ig-bound and total stool microbiota shared a subset of taxa (**A**), but at different abundances (**B**). Moreover, the Ig-bound and total stool microbiota grouped separately from the circulating cfmDNA (**A**,**B**). PD-1—programmed cell death protein 1.

**Figure 8 ijms-26-08063-f008:**
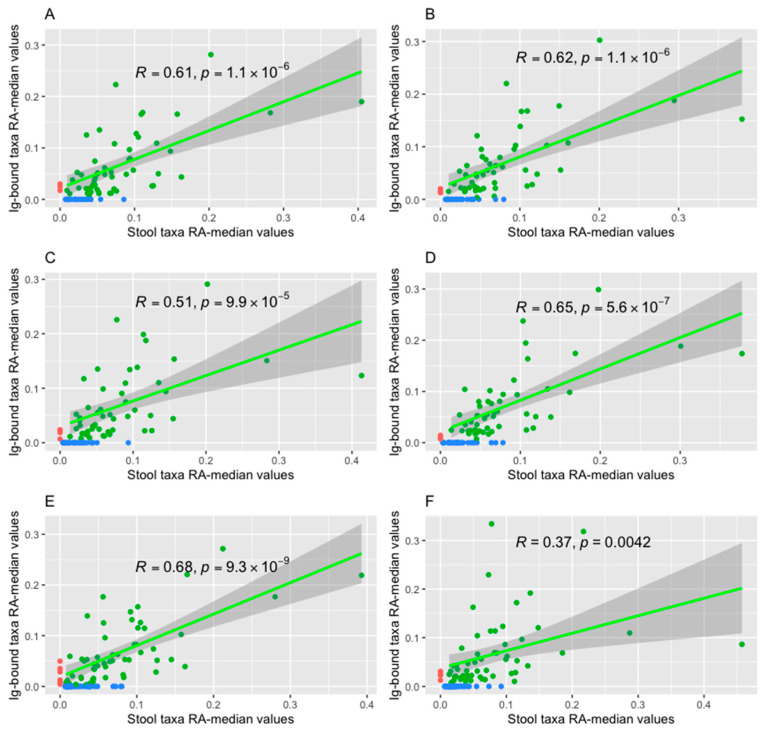
The correlations between the relative abundance (RA) median values of bacteria (at the genus level) detected in the immunoglobulin (Ig)-bound stool microbiota fraction and total stool microbiota (green dots and line) were analyzed in all advanced melanoma patients undergoing anti-PD-1 therapy (**A**,**B**), and subgroups of patients with clinical benefit—CB (**C**,**D**) or no clinical benefit—NB (**E**,**F**) from the immunotherapy, before its start—at T_0_ (0; (**A**,**C**,**E**)) and during treatment—at T_1_ (1; (**B**,**D**,**F**)). Statistics were performed using Spearman’s Rank Correlation Test. The *p*-value ≤ 0.05 was regarded as statistically significant. (**A**,**B**) show the trends observed in all patients at T_0_ and T_1_, respectively; (**C**,**D**)—in the CB0 and CB1 subgroups, respectively; (**E**,**F**)—in the NB0 and NB1 subgroups, respectively. Genera detected only in the total stool microbiota (blue dots) or in the Ig-bound stool microbiota fraction (red dots) were also indicated in the figures. The figures demonstrate positive correlations between the RA median values of genera in the Ig-bound stool microbiota and total stool microbiota in all advanced melanoma patients undergoing anti-PD-1 therapy, and the CB and NB subgroups at T_0_ and T_1_. In patients with favorable clinical outcomes, there was an increase in Spearman’s ρ value during treatment, while in those with unfavorable ones, there was an opposite trend. PD-1—programmed cell death protein 1.

## Data Availability

The data that support the findings will be available in the Repository for Open Data at https://doi.org/10.18150/6F2OSE following an embargo from the date of publication to allow for commercialization of research findings.

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
