# Peer review of "Characteristics of Intestinal Barrier State and Immunoglobulin-Bound Fraction of Stool Microbiota in Advanced Melanoma Patients Undergoing Anti-PD-1 Therapy"

_ijms, 2025, doi:10.3390/ijms26168063_

Round 1
Reviewer 1 Report
Comments and Suggestions for Authors
Dear Authors,
In the manuscript "Characteristics of intestinal barrier state and immunoglobulin-bound fraction of stool microbiota in advanced melanoma patients undergoing anti-PD-1 therapy" the authors investigated the relationship between intestinal barrier functionality and clinical outcomes of anti-PD-1 therapy in patients with advanced melanoma.
The manuscript presents a theme of significant scientific relevance. However, I have only a few comments.
- In the abstract, the authors indicated "anti-PD-1 therapy for patients with advanced melanoma." What does PD-1 mean?
- What about ELISA? It stands for Enzyme-Linked Immunosorbent Assay? Could you please describe it?
- What is the reason the authors did not include patients without a diagnosis for the proposed therapy?
- Were all 64 patients involved in the study men or women?
- In relation to the diet of patients receiving treatment, were they carnivorous or vegetarian?
- Did the patients use oral antimicrobials during the study?
- What was the criterion for excluding studies?
- I suggest that the authors include the catalog numbers of the kits, along with the reagents used in the study.
- What was the dilution factor of the antibodies used in the study?
Author Response
Comment 1: In the abstract, the authors indicated "anti-PD-1 therapy for patients with advanced melanoma." What does PD-1 mean?
Response 1: Thank you for your comment. The explanation of the abbreviation was added in the abstract (lines 18-19).
Comment 2: What about ELISA? It stands for Enzyme-Linked Immunosorbent Assay? Could you please describe it?
Response 2: Thank you for your comment. The explanation of the abbreviation was added in the abstract (lines 27-28).
Comment 3: What is the reason the authors did not include patients without a diagnosis for the proposed therapy?
Response 3: Thank you for your comment. Healthy subjects who served as controls were recruited to the study. However, due to the low number of individuals in the control cohort (n=10) and observed differences between patients and controls (e.g., in terms of age; controls were significantly younger than patients), we decided not to include them in our study. Previous studies analyzing the association between the gut microbiota and clinical outcomes of ICI in cancer patients also did not include controls. The primary goal of our study was to identify differences between responders and non-responders to assess the value of gut microbiota as a biomarker and target for enhancing responsiveness to treatment.
Comment 4: Were all 64 patients involved in the study men or women?
Comment 5: In relation to the diet of patients receiving treatment, were they carnivorous or vegetarian?
Comment 6: Did the patients use oral antimicrobials during the study?
Responses 4-6: Thank you for your comments. We have added a reference to our previous paper, in which the baseline characteristics of the study cohort, including sex distribution, dietary variables, and antibiotic use, were described (lines 129-134). There were no statistically significant differences in terms of sex between study subgroups. The majority of patients declared no antibiotic use up to 12 months before the start of the immunotherapy (we do not have information about the use of antimicrobials during the study). There were some differences in the dietary patterns between responders and non-responders (majority of them were carnivorous), in terms of plant portion consumption, prevailing dietary fat type, and dairy product consumption; however, we did not find any statistically significant association between diet and the intestinal barrier state biomarkers and we did not include these results in the study.
Comment 7: What was the criterion for excluding studies?
Response 7: Thank you for your comment. Patients were recruited to the study according to the criteria demonstrated in the Ministry of Health (Poland) drug program (reference 54) for treatment with anti-PD-1 therapy.
Comment 8: I suggest that the authors include the catalog numbers of the kits, along with the reagents used in the study.
Response 8: Thank you for your comment. Catalog numbers were added.
Comment 9: What was the dilution factor of the antibodies used in the study?
Response 9: The dilution factor of antibodies used for magnetic beads coating was 1:50.
Reviewer 2 Report
Comments and Suggestions for Authors
1. Can you wirte the sepcific purpose of the manuscript in part 1. Thank you. It can be written in the beginning of the sentence "the aim of this study is...".
2. Line 121 to 122 can be written in the disscusion part, thank you. This sentence seems like a conclusion or disscusion.
3. From line 140 to 144, the format of the typeface is wrong, please revise. Thank you.
4. Line 200, 251, the typeface is in the wrong format, please revice. Thank you.
5. The title of 3.1.1 was wrote in the wrong format, please change. Thank you. Same as the other third-level titles.
6. From line 331 to 335, the indent of paragraph is in a wrong format.
7. The article mentions that PD-1/PD-L signaling plays an important role in regulating IgA production and cites studies in mouse models by Kawamoto et al. and Zhang et al. However, these findings are from animal models and lack direct evidence in human patients. Although studies in animal models provide some reference for mechanism exploration, there are differences in physiology and immune responses between humans and animals, and relying only on animal results to extrapolate what happens in humans may somewhat undermine the persuasiveness of the argument; are there any current examples in humans?
8. The discussion suggests that SIgA may regulate intestinal microecology by affecting microbial function rather than just abundance, citing a study by Rollenske et al. which showed that binding of monoclonal IgA to bacterial antigens can lead to alterations in microbial function and metabolism, but the study was conducted on E. coli in a mouse model. In the human gut, where the microbial community is much more complex and diverse, and where there may be differences in the response of different bacteria to SIgA, can the adequacy of the evidence be strengthened by using the results of a single bacterial study in mice to support the argument in the complex intestinal environment of humans?
9. Can you write tables to demonstrate the outcomes of the indicators, thank you.
10. Please avoid using "our" in the manuscript, thank you.
11. Studies have found that high baseline fecal zonulin levels are associated with better survival outcomes, and it is discussed that this may be related to the role of zonulin in maintaining intestinal mucosal homeostasis. However, zonulin is often considered a marker of intestinal permeability, and elevated levels may be associated with impaired intestinal barrier function. However, the article is relatively simple in explaining why elevated zonulin was associated with positive clinical outcomes in this study. Can the possible mechanisms be explored in depth?
Author Response
Dear Reviewer,
Thank you very much for taking the time to review this manuscript. Please find the detailed responses below, along with the corresponding revisions in track changes in the resubmitted file.
Comment 1: Can you write the specific purpose of the manuscript in part 1. Thank you. It can be written in the beginning of the sentence "the aim of this study is...".
Response 1: Thank you for your comment. Our study aimed to analyze the association between the intestinal barrier state and clinical outcomes of anti-PD-1 therapy in advanced melanoma patients, and the introduction contains such information (lines 73-75).
Comment 2: Line 121 to 122 can be written in the disscusion part, thank you. This sentence seems like a conclusion or disscusion.
Response 2: Thank you for your comment. We have removed this sentence from the introduction.
Comment 3: From line 140 to 144, the format of the typeface is wrong, please revise. Thank you.
Comment 4: Line 200, 251, the typeface is in the wrong format, please revice. Thank you.
Responses 3-4: Thank you for your remarks. The errors were corrected.
Comment 5: The title of 3.1.1 was wrote in the wrong format, please change. Thank you. Same as the other third-level titles.
Response 5: Thank you for your comment. The third-level titles were written with the MDPI_2.3_heading3 format according to the MDPI template.
Comment 6: From line 331 to 335, the indent of paragraph is in a wrong format.
Response 6: Thank you for your comment. The errors were corrected.
Comment 7: The article mentions that PD-1/PD-L signaling plays an important role in regulating IgA production and cites studies in mouse models by Kawamoto et al. and Zhang et al. However, these findings are from animal models and lack direct evidence in human patients. Although studies in animal models provide some reference for mechanism exploration, there are differences in physiology and immune responses between humans and animals, and relying only on animal results to extrapolate what happens in humans may somewhat undermine the persuasiveness of the argument; are there any current examples in humans
Comment 8: The discussion suggests that SIgA may regulate intestinal microecology by affecting microbial function rather than just abundance, citing a study by Rollenske et al. which showed that binding of monoclonal IgA to bacterial antigens can lead to alterations in microbial function and metabolism, but the study was conducted on E. coli in a mouse model. In the human gut, where the microbial community is much more complex and diverse, and where there may be differences in the response of different bacteria to SIgA, can the adequacy of the evidence be strengthened by using the results of a single bacterial study in mice to support the argument in the complex intestinal environment of humans?
Response 7-8: Thank you for your valuable comments. To our knowledge, there is a lack of studies on humans that analyze the association between the SIgA production in the intestines and anti-PD-1 therapy. Moreover, the study of Rollenske et al. is the only one that analyzed the influence of IgA coating of bacteria on their functional potential. We are aware of the complexity and existing differences in the structure and physiology of the GI as well as the immune system between rodents and humans. However, it can be expected that the mechanisms that regulate the interaction between the gut microbiota and the host are similar. We have added such a comment to underline that the findings come from a preliminary study on a murine model (lines 906-909).
Comment 9: Can you write tables to demonstrate the outcomes of the indicators, thank you.
Response 9: Thank you for your comment. We have added such a table in the supplementary material (Table S3).
Comment 10: Please avoid using "our" in the manuscript, thank you.
Response 10: Thank you for your comment. Corrected.
Comment 11: Studies have found that high baseline fecal zonulin levels are associated with better survival outcomes, and it is discussed that this may be related to the role of zonulin in maintaining intestinal mucosal homeostasis. However, zonulin is often considered a marker of intestinal permeability, and elevated levels may be associated with impaired intestinal barrier function. However, the article is relatively simple in explaining why elevated zonulin was associated with positive clinical outcomes in this study. Can the possible mechanisms be explored in depth?
Response 11: Thank you for your comment. We hypothesized that increased permeability of the intestinal barrier may be associated with the bacterial antigens and metabolites translocation to the host milieu, where they may promote differentiation of immune cells that may migrate to the tumor microenvironment and induce anti-cancer responses. Moreover, there is also a hypothesis that bacterial antigens exhibit molecular similarity to tumor cells and therefore, induce anti-cancer immune responses. On the other hand, intestinal biomarkers, such as zonulin, may only reflect the immunological status (activated immune responses) of patients that favor the outcomes of the anti-PD-1 therapy (lines 663-674, 726-741).
Reviewer 3 Report
Comments and Suggestions for Authors
The publication Characteristics of intestinal barrier state and immunoglobulin-bound fraction of stool microbiota in advanced melanoma patients undergoing anti-PD-1 therapy analyzed stool samples from melanoma patients undergoing anti-PD-1 therapy for markers of intestinal integrity and inflammation, as well as for the composition of the microbiome.
Strengths of the study include the molecular techniques employed, the relatively high number of cases, and the repeated sampling before and during therapy.
However, the large number of subgroups and variables, particularly concerning the stool microbiome, warrants discussion, as it limits the statistical power of the findings.
In my view, a key point of criticism is that the authors allow themselves to interpret the associations between intestinal integrity, inflammation, and the microbiome with clinical therapy response as indications of causality. Furthermore, they fail to provide clinical data that would help elucidate the question of causality more convincingly.
The authors should not conclude that “The association between intestinal barrier state biomarkers, in particular faecal SIgA, and clinical outcomes of anti-PD-1 therapy in the study cohort revealed an important role of intestinal barrier functionality in shaping anti-cancer immune responses in advanced melanoma patients undergoing anti-PD-1 therapy,” but should instead restrict themselves to reporting the observed associations in the conclusions and abstract.
In the discussion, that could be shortened, the authors could discuss the alternative possibility that the intestinal inflammation and microbiome only reflect the immune activation and the efficacy of the patient's immune system.
It is evident that advanced malignant melanoma activates the patient’s immune system and that patients with a functional immune system are more likely to respond favorably to immune checkpoint inhibitor therapy than those who are immunosuppressed or whose immune system has been exhausted by the disease. Moreover, there is clinical evidence that melanoma patients who develop severe colitis requiring intense and prolonged therapeutic immunosuppression are unlikely to benefit from immunotherapy. It is highly probable that the relationships between immune activation, intestinal integrity, and the stool microbiome—as well as their interplay with additional immune stimulation by anti-PD-1 treatment, clinical response, and overall survival—are not linear but exhibit various optima.
High levels of SIgA and zonulin may simply reflect general melanoma-induced immune activation without indicating significant intestinal inflammation, while elevated calprotectin levels before treatment might reflect a less favorable immunologic state in terms of anti-cancer responsiveness or a predisposition to developing treatment-induced colitis.
Conversely, low levels of SIgA and zonulin could reflect immune exhaustion or pre-existing immunosuppression, both of which may negatively impact therapeutic outcomes.
To better distinguish between causation and association, the authors should provide additional clinical data and compare these with the stool analyses. Relevant data would include clinical stage, tumor burden (e.g., LDH levels), markers of general immune activation (e.g., CRP, lymphocyte, monocyte, and eosinophil counts), the occurrence of colitis, other immune-related side effects requiring immunosuppressive therapy, and the actual use of immunosuppressive drugs.
Author Response
Dear Reviewer,
Thank you very much for taking the time to review this manuscript. Please find the detailed responses below and the corresponding revisions in track changes in the re-submitted file.
Comment 1: In my view, a key point of criticism is that the authors allow themselves to interpret the associations between intestinal integrity, inflammation, and the microbiome with clinical therapy response as indications of causality. Furthermore, they fail to provide clinical data that would help elucidate the question of causality more convincingly.
The authors should not conclude that “The association between intestinal barrier state biomarkers, in particular faecal SIgA, and clinical outcomes of anti-PD-1 therapy in the study cohort revealed an important role of intestinal barrier functionality in shaping anti-cancer immune responses in advanced melanoma patients undergoing anti-PD-1 therapy,” but should instead restrict themselves to reporting the observed associations in the conclusions and abstract.
Response 1: Thank you for your valuable comments. We have modified the abstract (19-22), introduction (lines 121-125), summaries of the following paragraphs in the results (lines 203-205, 228-231, 287-301, 379-384, 455-457, 537-540, 642-644), discussion (lines 846-861), and conclusions (lines 1063-1079) to accurately report the observed associations, rather than indicating causality.
Comment 2: In the discussion, that could be shortened, the authors could discuss the alternative possibility that the intestinal inflammation and microbiome only reflect the immune activation and the efficacy of the patient's immune system.
Response 2: Thank you for your valuable comments. We have discussed an alternative possibility that trends observed in the functioning of the intestinal barrier in patients with favorable and unfavorable clinical outcomes could also simply mirror the overall immune activation and ability of patients’ immune system to fight cancer (lines 726-741).
Comment 3: To better distinguish between causation and association, the authors should provide additional clinical data and compare these with the stool analyses. Relevant data would include clinical stage, tumor burden (e.g., LDH levels), markers of general immune activation (e.g., CRP, lymphocyte, monocyte, and eosinophil counts), the occurrence of colitis, other immune-related side effects requiring immunosuppressive therapy, and the actual use of immunosuppressive drugs.
Response 3: Thank you for your valuable comment. We agree that a comparison between relevant clinical data and stool analyses would provide valuable insight into the paper. However, as you have noticed, the current version of the manuscript is large and the addition of new data would demand the extension of results and discussion, which, in our opinion, would make it difficult to follow by the readers. Clinical data, such as tumor stage, LDH concentration, and MDSC counts, were demonstrated in our previous papers DOI: 10.3390/cells12050789 and 10.3390/cancers14215369.
Round 2
Reviewer 2 Report
Comments and Suggestions for Authors
ok
Author Response
Dear Rewiever, Thank you for your valuable insight and acceptance of the revised manuscript.Reviewer 3 Report
Comments and Suggestions for Authors
I, myself, suggested that elevated calprotectin before treatment might be associated with negative treatment outcome as these patients might develop intense colitis that necessitates prolonged immunosuppression that could negatively impact treatment response. Nevertheless, the literature suggests that treatment induced colitis improves overall survival. That contradiction is the reason why calprotectin levels and overall survival may not be analysed without knowledge of factors like the actual development of colitis or other immune related adverse effects and the use of steroids or other immunosuppressant in the study population. Without these data and additional data on the immune status like serum CRP as well as blood eosinophils, lymphocytes and monocytes, the results of the study may not be adequately valuated.
Author Response
Comment
I, myself, suggested that elevated calprotectin before treatment might be associated with negative treatment outcome as these patients might develop intense colitis that necessitates prolonged immunosuppression that could negatively impact treatment response. Nevertheless, the literature suggests that treatment induced colitis improves overall survival. That contradiction is the reason why calprotectin levels and overall survival may not be analysed without knowledge of factors like the actual development of colitis or other immune related adverse effects and the use of steroids or other immunosuppressant in the study population. Without these data and additional data on the immune status like serum CRP as well as blood eosinophils, lymphocytes and monocytes, the results of the study may not be adequately valuated.
Response
Thank you for your valuable comment. We have collected the data of the peripheral blood cell counts of the study cohort before treatment (neutrophil, lymphocyte, monocyte, eosinophil, basophil, and platelet counts) from the medical records. Unfortunately, serum CRP level was not measured in the study cohort either before or during therapy.
Firstly, the associations of baseline peripheral blood counts and ratios with clinical outcomes of the anti-PD-1 therapy were analyzed. However, there were no statistically significant differences in the absolute counts of neutrophils (ANC), lymphocytes (ALC), monocytes (AMC), eosinophils (AEC), basophils (ABC), and platelets (PLT), and neutrophil to lymphocyte (NLR), monocyte to lymphocyte (MLR), and platelet to lymphocyte ratios (PLR) in the periphery between R vs. NR and CB vs. NB subgroups (p-value > 0.05, Table S3). Subsequently, the peripheral blood cell counts and ratios were compared between subgroups of patients with low vs. high concentrations of fecal zonulin, SIgA, and calprotectin at baseline (the cut-off points of intestinal barrier state biomarkers indicated in the survival analysis were used for patient’s categorization to appropriate subgroup). This analysis indicated that the median level of peripheral ABC was higher in patients with fecal zonulin level higher than 303 ng·mL-1 as compared to those with fecal zonulin level lower or equal to 303 ng·mL-1 (50/µL vs. 30/µL p-value = 0.01, Table S3).
Moreover, the correlations of peripheral blood cell counts and ratios with the levels of intestinal barrier state biomarkers at baseline were also investigated. There were found several moderate, but statistically significant correlations between analyzed biomarkers; however, they were distinct in patients with favorable and unfavorable clinical outcomes (p-value ≤ 0.05, Table S4). In detail, analysis of all samples indicated negative correlations between fecal calprotectin and peripheral ALC, ABC, and PLT, and between fecal zonulin and peripheral ABC. In contrast, fecal calprotectin positively correlated with MLR in the periphery in the study cohort.
In patients with favorable clinical outcomes, i.e., R and CB subgroups, fecal calprotectin and zonulin negatively correlated with peripheral PLT. Additionally, fecal calprotectin negatively correlated with peripheral ABC in these subgroups.
On the other hand, in the NB subgroup, fecal zonulin positively correlated with peripheral ANC and NLR, while in the NR subgroup, fecal SIgA positively correlated with peripheral ABC and fecal calprotectin with peripheral MLR.
The analysis indicated the correlation between intestinal barrier state biomarkers and peripheral blood cell counts and ratios; however, it does not provide answer to the question whether indicators of intestinal barrier functionality simply reflect the immune status of patients. Future mechanistic studies are required to elucidate this aspect as peripheral blood cell counts do not fully reflect the functionality of these cells or immune cell infiltration into the tumor microenvironment.
Results of the analysis are presented in the Supplementary Materials (Table S3 and S4) and main findings are shortly described in the discussion (lines 728-744).
Furthermore, we analyzed the association between fecal calprotectin (associated with higher risk of progression and death) and development of irAEs in the study cohort. However, a small subset of patients developed irAEs (13/58; 22%), which is in line with trends observed in other cancer cohorts treated with the anti-PD-1 therapy (DOI: 10.1038/ s41571-019-0218-0), and only 31% (4/13) of these patients were using steroids. All of patients, who developed irAEs, responded to the anti-PD-1 therapy. We did not find any statistically significant differences in the baseline fecal calprotectin level between patients, who developed irAEs vs. those, who did not; however, baseline fecal calprotectin level tended to be lower in patients, who developed irAEs (even high grade irAEs ≥ 3) than in those, who did not. Therefore, it cannot be hypothesized that high fecal calprotectin level may reflect patient predisposition to develop irAEs/ICI-induced colitis.
|
|
irAEs (No) |
irAEs (Yes) |
p-value |
|
Calprotectin Median (range) |
81.67 (4.75-1491) |
61.41 (11.21-521.9) |
0.5646833 |
|
|
No irAEs |
G1 |
G2 |
G3 |
p-value |
|
Calprotectin Median (range) |
81.67 (4.75-1491) |
39.68 (18.28-211.9) |
136.8 (61.41-521.9) |
12.01 (11.21-12.81) |
0.08364 |
|
|
No irAEs |
Low (G<3) |
High (G≥3) |
p-value |
|
Calprotectin Median (range) |
81.67 (4.75-1491) |
100.9 (18.28-521.9) |
12.01 (11.21-12.81) |
0.08183 |
Round 3
Reviewer 3 Report
Comments and Suggestions for Authors
The additional data are very useful for interpretation of the results.